# Halogenation of tyrosine perturbs large-scale protein self-organization

Huan Sun [1,2], Haiyang Jia [1,3] ✉, Olivia Kendall[2,4], Jovan Dragelj[2], Vladimir Kubyshkin[5], Tobias Baumann [2], Maria-Andrea Mroginski [2] ✉, Petra Schwille [3] ✉ & Nediljko Budisa [2,5] ✉

Protein halogenation is a common non-enzymatic post-translational modification contributing to aging, oxidative stress-related diseases and cancer. Here, we report a genetically encodable halogenation of tyrosine residues in a reconstituted prokaryotic filamentous cell-division protein (FtsZ) as a platform to elucidate the implications of halogenation that can be extrapolated to living systems of much higher complexity. We show how single halogenations can fine-tune protein structures and dynamics of FtsZ with subtle perturbations collectively amplified by the process of FtsZ self-organization. Based on experiments and theories, we have gained valuable insights into the mechanism of halogen influence. The bending of FtsZ structures occurs by affecting surface charges and internal domain distances and is reflected in the decline of GTPase activities by reducing GTP binding energy during polymerization. Our results point to a better understanding of the physiological and pathological effects of protein halogenation and may contribute to the development of potential diagnostic tools.

The lifespan of living cells and organisms is influenced by the quality of their proteins. The primary sequence of a protein is one of the most important determinants of protein folding and final conformation as well as biochemical activity, stability, and half-life. Enzymatic and non-enzymatic post-translational modifications (PTMs) greatly expand the structural and functional space of proteins, as well as the proteome as a whole. Many protein PTMs produce the non-enzymatic attachment of specific chemical groups to amino acid side chains, e.g., glycation, nitrosylation, oxidation/reduction, acetylation, succination, and halogenation. From a structural perspective, these modifications usually alter the overall tertiary fold, often destabilize proteins, and may result in cleavage of the protein backbone or protein aggregation[1]. Such a process is considered as protein damage and has a particularly pronounced deteriorating impact in the quality of individual proteins, but also on protein complexes and cellular activities that require protein self-assembly.

Halogenation is a typical pathological modification that can occur in the context of oxidative stress in the environment. Oxidative halogenation has been shown to impair proper cellular functions and cause severe long-term health problems such as aging, cancer, or even death[2-4]. The aromatic amino acid tyrosine is a major target for oxidative halogenation in proteins. Halogenation at this residue can occur endogenously; for example, hypochlorous acid generated by the myeloperoxidase-hydrogen peroxide-chloride system of phagocytes has been shown to be the major source of 3-chlorotyrosine formation in cells and tissues[5-7]. In addition, the widespread use of halogenated aromatic compounds in agriculture, dye, chemical and pharmaceutical industries[8,9] raises major concerns about acquired halogenation, as reactive halogen species might be the direct source of halogenation of both enzymatic and non-enzymatic origin[10]. For this reason, the study of halogen modifications and their role in protein-based systems is of enormous importance in understanding the cellular mechanisms of

[1]School of Chemistry and Chemical Engineering, Beijing Institute of Technology, Beijing 100081, PR China. [2]Technical University of Berlin, Müller-Breslau-Str. 10, D-10623 Berlin, Germany. [3]Max Planck Institute of Biochemistry, Am Klopferspitz 18, D-82152 Martinsried, Germany. [4]University of Edinburgh, David Brewster Road, King's Buildings, Edinburgh EH9 3FJ, UK. [5]University of Manitoba, 144 Dysart Rd., R3T 2N2 Winnipeg, MB, Canada. ✉e-mail: oceanjia0821@gmail.com; andrea.mroginski@tu-berlin.de; schwille@biochem.mpg.de; nediljko.budisa@umanitoba.ca

oxidative damage. Such information provides insights that might lead to novel strategies in biomedical approaches that prevent and heal proteome damage.

The emerging technology of genetic code expansion has enabled the co-translational and site-specific incorporation of diverse non-canonical amino acids (ncAAs) that confer versatile physicochemical and biological properties to proteins[11,12]. Genetic introduction of selective chemical modifications into a protein of interest provides a powerful approach to characterize the structural and biochemical consequences of the modifications[13,14]. By mimicking PTMs, the site-specific incorporation of the ncAAs can provide information on how the position, density, and distribution of protein modifications perturb protein structures and functions on a small scale. Co-translational modification of target proteins with oxidized ncAAs at defined positions has already proven to be a useful tool to study the role of protein nitration[15] or oxidation[16] at specific positions. For example, nitrotyrosine has been genetically incorporated in superoxide dismutase from mitochondria to encode the protein oxidative damage[17]. The effect of protein modifications in collective intramolecular processes[18] in protein complexes assembly[19,20] and in cells and tissues[15,21,22] have also been studied. For instance, modifications such as acetylation[20] and methylation[19] have been used to determine the effects of modifications on nucleosome complexes assembly and cellular transcriptional responses. Nonetheless, there is still no insight into how oxidative modifications could act on large scale protein assemblies. In particular, the amplification effects during maturation of complex structures are still unclear. The high cellular noise and the extreme complexity of the system under study pose a major challenge.

To address this knowledge gap, we here employ a reconstituted bacterial minimal system of the cytoskeleton to gain insight into the effects of halogenation on the dynamic process of molecular self-organization. Since cytoskeleton components are major actors in the cellular lifecycle, a perturbation or disruption of its functions would contribute significantly to aging[5,23]. The minimal system studied here[24] is a one-component biological system reconstituted in vitro from scratch with the purified FtsZ protein (the *ftsZ* gene product), the known prokaryotic homologue of the eukaryotic protein tubulin (Fig. 1a). The protein has been shown to be sensitive to halogenating chemicals[25] and its activity is sensitive to modifications at individual sites[26]. As an essential part of the bacterial division ring, known as "Z ring", FtsZ has shown intriguing self-organization when reconstituted in vitro on biological membranes. It polymerizes into dynamic vortices by circular treadmilling dynamics fueled by GTP hydrolysis[27]. These properties of FtsZ make it a promising candidate for studying the collective behaviors of PTMs in vitro. By combining experiments and theory, we demonstrate how incorporation of halogenated tyrosine analogues (HYs) at a key position of FtsZ perturbs its structural features and GTPase activity. Consequently, the precise and highly tailored perturbations of structures and dynamics can be collectively amplified through protein self-organization (Fig. 1b). Finally, we could elucidate the detrimental effects of halogenated tyrosine on collective FtsZ self-organization behavior in the reconstituted minimal cell division system and further interpret how the change at a single site poisons the system globally.

## Results

### Co-translational incorporation of halogenated tyrosine analogues

In living systems, oxidation of tyrosine residue can often lead to the formation of halogenated residues known as HYs[3]: 3-chloro-tyrosine (ClY), 3-bromo-tyrosine (BrY), 3-iodo-tyrosine (IY), 3,5-dichloro-tyrosine (Cl$_2$Y), 3,5-dibromo-tyrosine (Br$_2$Y), and 3,5-diiodo-tyrosine (I$_2$Y)[28–31]. Halogenation affects several key molecular properties of the parent residue: molecular volume, p$K_a$ of the side-chain group, and hydrophobicity (Fig. 1c). We studied the experimental p$K_a$ values of the

phenolic group in free amino acids, and found that the value depends primarily on the number of halogen atoms introduced to the molecule: Tyr (p$K_a$ 9.9) > ClY ≈ BrY ≈ IY (p$K_a$ 8.3) > Cl$_2$Y ≈ Br$_2$Y ≈ I$_2$Y (p$K_a$ 6.5). Next, we examined the hydrophobicity of the amino acid residues using lipophilicity measurements using a recently developed method[32]. The evaluation of the residue lipophilicities is complicated because the ionization of the side chain is also perturbed by halogenation. Thus, we examined the experimental distribution coefficient (logD) in a range of pH values (pH 6–9) to illustrate possible changes in the ionization state. We found that the experimental lipophilicity values led to a rather complex outcome: while each newly introduced halogen atom increased the hydrophobicity of the side-chain, the concomitant decrease of p$K_a$ facilitated the transition to the deprotonated form, and thus lowered the hydrophobicity. This complex interplay between halogenation and the properties of the tyrosine residue makes it rather difficult to make accurate predictions without analyzing the actual experimental protein data. Since the microenvironment of the protein affects the p$K_a$ transition of the tyrosine side chain, halogenation may reduce overall hydrophobicity in one position, where p$K_a$ falls below the pH, but elevate the hydrophobicity at another position, where p$K_a$ stands above pH of the medium.

Thus, to examine the roles of modifications in proteins, we set out to produce corresponding proteins containing above-mentioned modifications and study their properties. We employed the stop codon suppression technique, in which so-called orthogonal pairs are used. An orthogonal pair consists of an aminoacyl-tRNA synthetase (aaRS), which selects the amino acid of interest from the pool of intracellular substances, and a transfer-RNA (tRNA), which carries the amino acid to the ribosome and incorporates it into the sequence in response to an in-frame stop codon. Few methods have been developed for incorporation of HYs into proteins[33–38], however, none of them was sufficient to incorporate the full set of analogues selected for this study because they are typically being restricted to 3-halo-tyrosines only.

To selectively incorporate the HYs into the protein of interest, we constructed a gene library based on *Methanocaldococcus jannaschii* TyrRS (*Mj*TyrRS) by randomizing eight active-site residues of the aminoacyl-tRNA synthetase (detailed mutations see Supplementary Table 1). The *Mj*TyrRS library was then selected by one-round positive selection[14,39] (Supplementary Fig. 1) against HYs. For selection, we used chloramphenicol acetyltransferase with two amber codons and a small ubiquitin-like modifier tagged superfolder green fluorescent protein (sfGFP) with one in-frame amber stop codon for readout. After a single round selection, the *Mj*TyrRS mutant B48RS (Y32G, L65E, H70G, F108Q, Q109C, D158A, L162N) was selected as a candidate with high selectivity towards several HYs: ClY, Cl$_2$Y, BrY, Br$_2$Y, and IY. Another mutant C64RS (Y32V, L65E, H70G, F108N, D158A, L162C) was selected for incorporation of I$_2$Y (Supplementary Fig. 2). Incorporation of HYs into sfGFP using the mutants B48RS (for ClY, Cl$_2$Y, BrY, Br$_2$Y, and IY) and C64RS (for I$_2$Y) was also confirmed by mass spectra of the intact protein obtained after electrospray ionization (Supplementary Fig. 3). Incorporation efficiency was also assessed by comparing the fluorescence intensity of sfGFP containing HYs (Supplementary Fig. 2). Markedly, in the expression experiments, the mutant B48RS showed an almost equal efficiency with Cl$_2$Y, BrY, Br$_2$Y, and IY, which in each case was more than 100-fold higher than the control without HYs additions (Supplementary Fig. 2).

### Single-site halogenation suppresses protein activity

After establishing the tool for HYs incorporation, we next prepared FtsZ variants containing the analogues. Within the native asymmetric unit FtsZ, it adopts two distinct conformational states[40–43] with its two subdomains acting independently. Therefore, positions localized at or near the interface may dramatically affect overall conformation transitions[44,45]. We incorporated ClY, Cl$_2$Y, BrY, Br$_2$Y, IY and I$_2$Y at

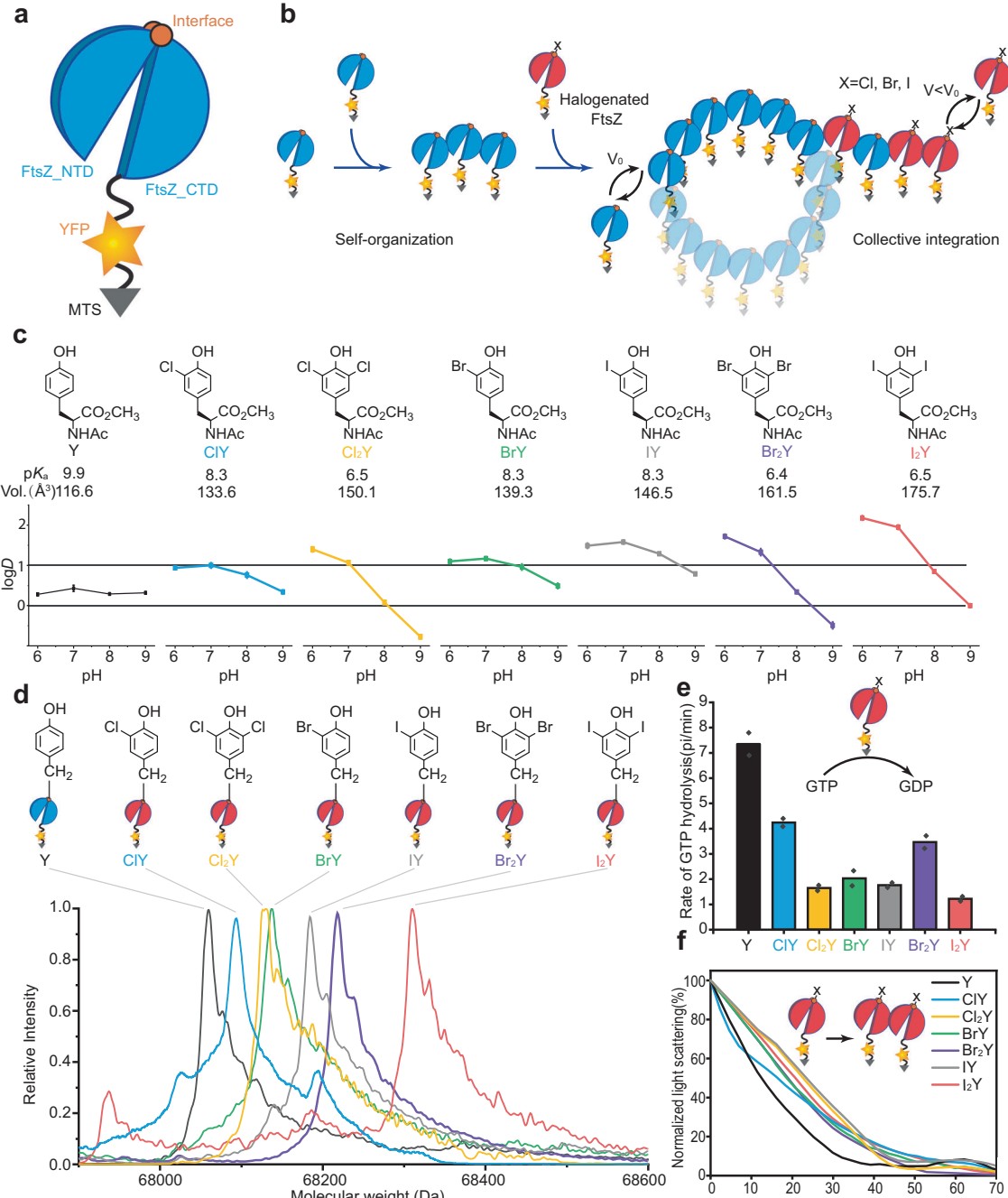

**Fig. 1 | Genetic incorporation of halogenated tyrosine analogues in FtsZ based constructs. a** Schematic illustration of the FtsZ construct (FtsZ-YFP-mts): FtsZ (1–366, split blue cylinder), yellow fluorescent protein (YFP, yellow star) and a membrane target sequence (mts) (grey triangle). V and $V_0$ indicate the treadmilling speed. **b** Schematic model for inhibition of FtsZ ring formation by halogenated residues. **c** Molecular properties of the tyrosine analogues as dependent from halogenation. Experimental lipophilicity was determined in methyl N-acetylaminoacetates against buffers at different pH values, p$K_a$ of the phenolic side-chain was experimentally determined in free amino acids, molecular volume is calculated for free amino acids. **d** Chemical structures of the halogenated tyrosine analogues incorporated into FtsZ and the deconvoluted ESI-MS spectra. The expected and observed masses are listed in Supplementary Table 3. **e** GTPase activities of wild type FtsZ-YFP-mts (Y) and halogenated FtsZ(Y222X)-YFP-mts (X = ClY, BrY, Br$_2$Y, I$_2$Y, Cl$_2$Y, and IY). Data from two independent replicates are shown as means with standard deviations. **f** Dynamic light scattering of wild type FtsZ-YFP-mts and halogenated FtsZ-YFP-mts variants. Source data of **c**–**f** are provided as a Source Data file.

position Tyr222, which is the only tyrosine structurally close to the boundary between the N-terminal domain and C-terminal domain. The position is located on the outer surface of the protein and is exposed to solvent, allowing us to mimic natural halogenation progress that is modified by external natural modifiers. In addition, this position is known to be sensitive to PTMs[26,46], making it a good candidate for investigation of halogenations.

To visualize the FtsZ dynamic patterns in vitro, we designed a truncated FtsZ(1-366) variant fused with yellow fluorescent protein (YFP) and a membrane-targeting sequence (mts)[47] (Fig. 1a). The FtsZ-YFP-mts variants were expressed in *E. coli* harboring plasmid-borne B48RS and C64RS orthogonal pairs in media supplemented with HYs. Fidelity of the HYs incorporation was verified by mass-spectra analysis (Fig. 1d), which demonstrated that the FtsZ protein isolates were

homogeneously labelled with HYs at position 222. Deconvoluted mass spectra revealed the masses expected for the wild type and ClY, Cl$_2$Y, BrY, Br$_2$Y, IY and I$_2$Y containing FtsZ-YFP-mts variants (Fig. 1d, Supplementary Fig. 4, Supplementary Table 3). The mass shifts caused by ClY, Cl$_2$Y, BrY, Br$_2$Y, IY and I$_2$Y were +33.31, +69.33, +78.40, +158.70, +125.33 or +251.88 Da, respectively, compared with the wild type protein. This result would be expected for the substitution of one or two hydrogen atoms with halogens at Tyr222.

To further evaluate the influence of the site-specific halogenation, we examined the GTPase activity and polymerization rate in the protein samples. Curiously, we found that even a single hydrogen-to-halogen atom exchange was sufficient to alter the enzymatic activity. Indeed, we found that GTPase activity was lowered in all cases by the presence of HYs. Particularly low activities were observed for Cl$_2$Y, IY, and I$_2$Y containing variants, which showed a decrease in GTP hydrolysis rate by up to 70% (Fig. 1e). In addition, light scattering results showed that all halogenated proteins exhibited suppressed polymerization rates compared with wild type (Fig. 1f). The decreased GTPase activity and polymerization rates may be caused by reduced GTP hydrolysis as a result of molecular perturbations produced by halogenation. We infer that the molecular perturbations could result from the changes of key molecular properties demonstrated in Fig. 1c, such as side-chain acidity, molecular volume, and hydrophobicity. As mentioned before, the p$K_a$ values of HYs at pH7.5 were all lower than that of tyrosine and the lipophilicity was higher than that of tyrosine, which could increase the acidity of tyrosine side-chain and hydrophobicity, leading to suppressed GTPase activities. In addition, halogenation can enlarge the volumes of the amino acids, and thus might disrupt the protein structures.

## Halogenations of tyrosine affect protein assembly and dynamics

We next examined the ring formation behavior of halogenated FtsZ in live *E. coli* cells (Supplementary Fig. 5) using fluorescence confocal microscopy. To this end, we examined microscope images of cells overexpressing the FtsZ-YFP-mts constructs. We observed that with the wild type protein, multiple complete rings were relatively evenly distributed along the cell length (Supplementary Fig. 5a), consistent with previous reports[48,49]. In contrast, the cells containing FtsZ(Y222ClY)-YFP-mts produced only a single ring in the center (Supplementary Fig. 5d). Conversely, the BrY and Br$_2$Y containing variants formed multiple interconnected short helices, densely distributed along the cells (Supplementary Fig. 5 b, e). FtsZ(Y222Cl$_2$Y)-YFP-mts and FtsZ(Y222IY)-YFP-mts produced incomplete bands or arcs that could be parts of helix or spiral and ring fragments (Supplementary Fig. 5c, f). Finally, no ring formation was observed in the cells overexpressing I$_2$Y modified protein (Supplementary Fig. 5g). Such severe changes of FtsZ architectures may be caused by the altered structures of FtsZ proteins during halogenations, which requires further structural analysis. To investigate the dynamics of FtsZ assembly in vivo, we performed fluorescence recovery after photobleaching (FRAP) analysis on these FtsZ architectures. In FRAP analysis, a portion of a fluorescent FtsZ pattern is photo-bleached with a focused laser beam and the recovery of the fluorescence signal caused by replacement of subunits outside the photobleached region is measured. We found that the fluorescence recovery of wild type FtsZ-YFP-mts was faster than that of the halogenated FtsZ-YFP-mts (Supplementary Fig. 5h–i), which is consistent with the trend of GTPase activity shown in Fig. 1e. All of the above results are sufficient to ascertain effects on both structures and dynamics of FtsZ resulting from single site halogenation. However, because of the physiological complexity and high background of endogenous FtsZ, it is not yet possible to deduce a specific underlying molecular mechanism and the specific effects of the different types of halogenations from these cellular results.

To better understand the architecture and dynamics of the filament network generated by halogenated FtsZ, we reconstituted the

protein variants on supported lipid bilayers (SLBs) and quantified their self-organized patterns using a total internal reflection fluorescence microscope (TIRFM) (Fig. 2a, Supplementary Movie 1). First, SLB with negatively charged lipid composition was prepared in a home-made microscope chamber. Then FtsZ was introduced on the SLB and self-assembly of FtsZ was initiated by addition of GTP. In our experiments, we found that the wild type FtsZ-YFP-mts could be recruited to the membrane, where it can promptly self-organize into a homogeneous treadmilling ring pattern (Fig. 2b). The variants containing ClY, BrY, Br$_2$Y, and I$_2$Y were also able to form rings, albeit with heterogenous ring morphology (Fig. 2b, c). The average diameters of the rings formed by FtsZ(Y222ClY)-YFP-mts (0.62 ± 0.15 μm) were 22.5% smaller than those formed by the wild type protein (0.80 ± 0.18 μm). However, in the case of FtsZ(Y222BrY)-YFP-mts, FtsZ(Y222Br$_2$Y)-YFP-mts and FtsZ(Y222I$_2$Y)-YFP-mts, the ring diameters were 11.3%, 15.0% and 28.8% larger than in the wild type, respectively. Moreover, FtsZ variants modified with closely analogues ClY and BrY, characterized with similar side-chain acidity and molecular volume (Fig. 1c), exhibited self-organization into significantly distinct sizes of ring patterns. This fact indicates that even subtle volume differences (6 Å$^3$) may be amplified by large-scale protein self-organization, resulting in 43.5% enlarged ring diameters. Overall, single atom modifications allowed us to precisely fine-tune the protein-assembly progress, and the changes are even visible through fluorescence microscopy.

In the following step, we examined the treadmilling dynamics[50] of FtsZ. Treadmilling is a GTPase dependent process that results from polymerization at one end and depolymerization at the other end of a polymer chain[51]. Compared with the more noisy and complex in vivo approaches (Supplementary Fig. 5), we expected treadmilling dynamics to provide a clearer illustration of the protein damage, since this assay is performed purely in vitro. Since the deficiency in the GTPase activity (Fig. 1d) suggests a strong perturbation of this dynamic process from halogenation, we quantified the vortices' velocities by the slope of kymographs[47] (Fig. 2d) generated along the circumference. Results showed lower velocities in all halogenated proteins, as expected. The halogenations decrease the mean rotation velocities down to 48.0% compared to the wild type FtsZ-YFP-mts (21.63 ± 5.11 nm s$^{-1}$) (Fig. 2e and Supplementary Table 4).

In two cases, the presence of HYs resulted in full suppression of the ring pattern formation in vitro. We did not observe any distinctive ring formation produced by protein variants containing Cl$_2$Y and IY. These proteins display fiber-like and highly meshed filament patterns in the entire membrane area (Fig. 2b). Then we analyzed the curvature of FtsZ filaments with stretching open active contours (SOAX)[52]. Compared to the wild type FtsZ-YFP-mts, FtsZ filaments containing Cl$_2$Y and IY displayed decreased curvatures indicating that the two types of halogenations may seriously bend the phenotype of FtsZ filaments and further inhibit the ring pattern formation (Fig. 2h). Instead of ring velocity, we analyzed their turnover rate by FRAP to better understand their assembly dynamics. With the wild type FtsZ-YFP-mts, a relatively fast recovery was observed with a half-time 4.97 ± 0.24 s (*n* = 6). Conversely, with Cl$_2$Y and IY containing proteins the half-time was nearly doubled, indicating slower dynamics (Fig. 2f, g and Supplementary Table 4). The lowered dynamics comes in agreement with the in vivo results (Supplementary Fig. 5h–i). To better understand the prominent changes exhibited by IY and Cl$_2$Y containing proteins, we compared the molecular properties of the amino acids. The side-chain acidity measured in free amino acids showed distinctly different p$K_a$ values to those of tyrosine: 9.9 in tyrosine, 8.3 in IY, and 6.5 in Cl$_2$Y. At the same time, the molecular volume of the analogues was very similar, differing by only 3.6 Å$^3$ (Fig. 1c). In addition, we compared the polarity of the side-chains in model compounds. The halogenated amino acids IY and Cl$_2$Y exhibited identical lipophilicity in non-ionized form (measured at pH 6) and both were more hydrophobic than any amino acid in the canonical protein repertoire (Fig. 1c).

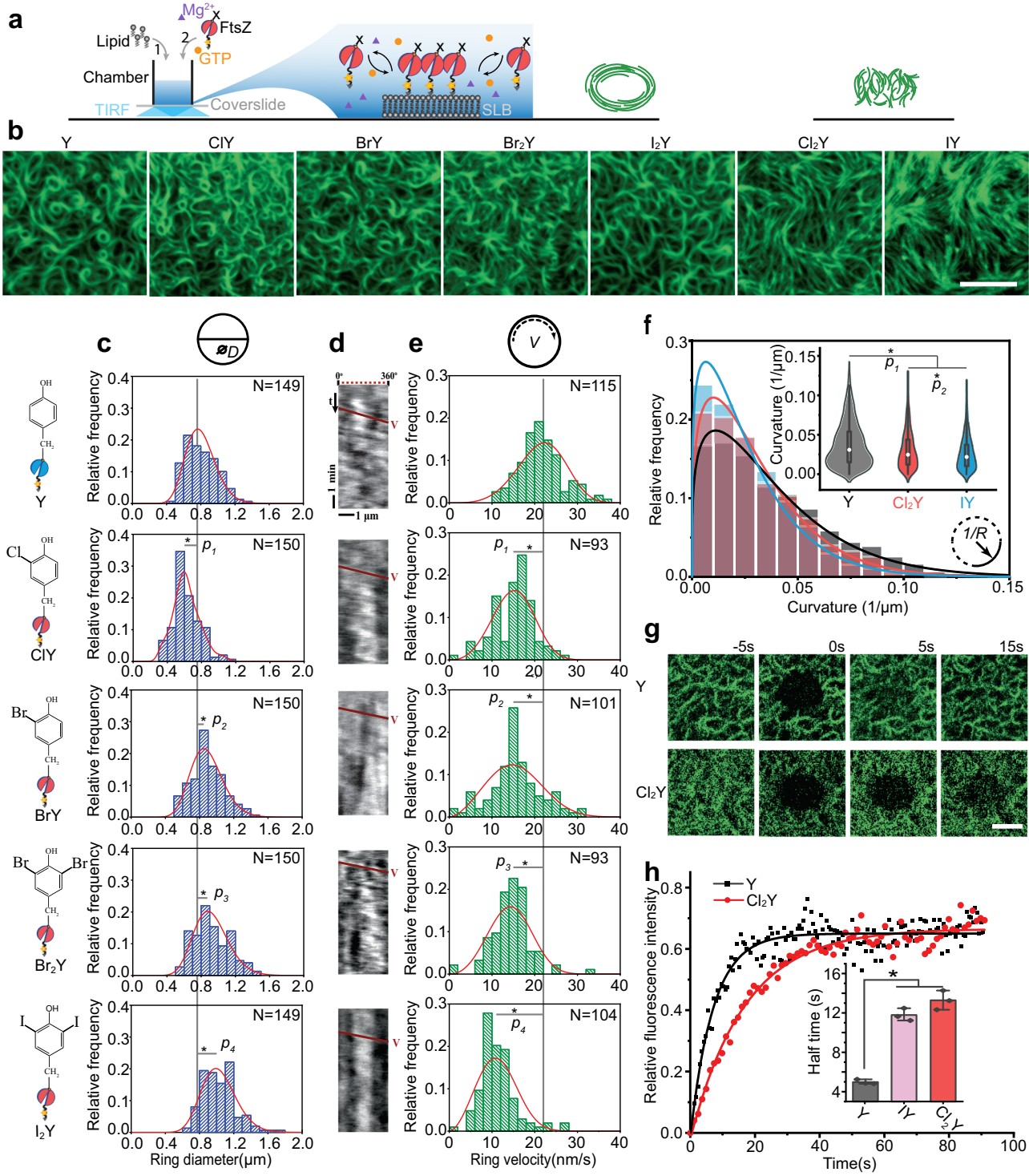

The high similarity in molecular volume and hydrophobicity of the side chains of IY and Cl$_2$Y may explain the similar behavior of the FtsZ-YFP-mts constructs with these residues. Overall, these results clearly indicate that halogenation not only perturbs enzymatic activity, but that the associated effects on structure and dynamics are clearly downstream of the self-organization process.

## Partial halogenation disrupts ring pattern formations

To investigate the collective progress of halogenated FtsZ integration in vitro, we performed a self-organization assay by mixing the FtsZ chimera bearing HYs with wild type protein. We selected two variants: the wild type as the one exhibiting native self-assembly and the Cl$_2$Y-containing protein as the one unable to form rings. The proteins were mixed in different proportions, with FtsZ(Y222Cl$_2$Y)-YFP-mts provided at 25%, 50%, and 75%, yielding a total protein concentration of 0.5 μM. The formation of FtsZ ring pattern was examined on SLB under TIRF-monitoring (Fig. 3a, Supplementary Movie 2). FtsZ architectures and dynamics, such as ring diameters, velocity and filament curvatures were analyzed. We observed that the density of the multiple homogeneously distributed rings decreased upon addition of the halogenated sample. The density values were 1.44 (±0.12) × 10$^5$ μm$^{-2}$, 1.01 (±0.04) × 10$^5$ μm$^{-2}$, and 5.28 (±0.21) × 10$^4$ μm$^{-2}$ upon the presence of 0%, 25%, and 75% of the Cl$_2$Y modified variant, respectively (Fig. 3b–c). By analyzing the ring morphology and treadmilling

**Fig. 2 | Halogenation affects FtsZ patterning and treadmilling dynamics.**
**a** Schematic illustrating FtsZ self-organization on model membrane monitored by TIRF. **b** Representative cytoskeletal pattern of wild type FtsZ-YFP-mts (Y) and halogenated FtsZ(Y222X)-YFP-mts (X = ClY, BrY, Br$_2$Y, I$_2$Y, ClY, and IY) on supported membrane (0.5 μM proteins, 4 mM GTP and 1 mM Mg$^{2+}$). Scale bar: 3 μm. The experiment was performed three times under identical conditions. **c** Ring size distributions of wild type FtsZ-YFP-mts and halogenated FtsZ-YFP-mts. D indicates the ring diameter. *Analysis of Variance (ANOVA) one-way statistical test ($p_1 = 6.16 \times 10^{-18}$; $p_2 = 4.91 \times 10^{-5}$, $p_3 = 1.28 \times 10^{-7}$; $p_4 = 1.18 \times 10^{-20}$). **d** Representative kymograph along the circumference of the vortices formed by wild type FtsZ-YFP-mts and halogenated FtsZ-YFP-mts. The respective slopes (red lines) correspond to the treadmilling velocity (V) of the vortices. **e** Velocity distributions for wild type FtsZ-YFP-mts and halogenated FtsZ-YFP-mts. *V* indicates the ring velocity. *Analysis of Variance (ANOVA) one-way statistical test ($p_1 = 9.57 \times 10^{-18}$; $p_2 = 2.01 \times 10^{-13}$,

$p_3 = 6.08 \times 10^{-21}$; $p_4 = 4.92 \times 10^{-39}$). **f** Filament curvature distributions for wild type FtsZ-YFP-mts ($n = 1700$ filaments) and FtsZ(Y222 Cl$_2$Y/IY)-YFP-mts ($n = 700$ filaments). Wild type FtsZ-YFP-mts (Y) is shown in black, FtsZ(Y222Cl$_2$Y)-YFP-mts (Cl$_2$Y) is red and FtsZ(Y222IY)-YFP-mts is blue (IY). *1/R* represents the filament curvature. *Analysis of Variance (ANOVA) one-way statistical test ($p_1 = 6.00 \times 10^{-25}$; $p_2 = 3.19 \times 10^{-4}$). Box plots in **f**: the lines represent medians, box limits represent quartiles 1 and 3, whiskers represent 1.5 × interquartile range and points are outliers. **g** Snapshots and **h** fluorescence recovery curves for wild type FtsZ-YFP-mts and FtsZ(Y222Cl$_2$Y)-YFP-mts after photo-bleaching. Scale bar: 3 μm. Inset: The half-life of fluorescence recovery. Data from three independent replicates are shown as means with standard deviations. *Analysis of Variance (ANOVA) one-way statistical test ($p_1 = 1.28 \times 10^{-5}$). The solid curves in **c** and **e** represent the Gaussian fit. The solid curves in **f** represent the extreme fit. Source data of **c**, **f** and **h** are provided as a Source Data file.

velocity, we found that the filament curvature gradually decreased upon addition of the halogenated protein. A gradual enlargement in ring sizes and a slowdown in ring dynamics were observed as well (Fig. 3d). Finally, the increased proportion of halogenated FtsZ resulted in self-organization patterns that became increasingly similar to those of pure Cl$_2$Y containing FtsZ, with a highly meshed filament pattern (Fig. 3b). These observations confirm that halogenation on FtsZ at Y222 position impairs the assembly of protofilaments from monomeric proteins and ultimately inhibits Z-ring formation.

### Structure modelling elucidates the effects of halogenation

To better understand how the subtle changes in halogenated protein structure suppress protein activity and how the effects are further amplified by the self-organization process, we constructed structural models for the wild type FtsZ monomer and dimer, as well as for the FtsZ(Y222Cl$_2$Y) mutant including a GTP molecule at the interface and in the upper monomer binding site. Starting coordinated were generated from a homology model of the wild type FtsZ with the I-TASSER suite[53] using the *M. jannaschii* FtsZ dimer structure (PDB code: 1W5A)[40] (Fig. 4a, Supplementary Fig. 6) as template. Compared to the published structures of the two monomer conformations[41], the structural models were identified to be in the closed state i. e., a conformation preferred in solution. Thus, the FtsZ dimer models represent the nucleation of a filament. Structures containing the mutated residues FtsZ(Y222Cl$_2$Y) were modelled by modifying the tyrosine residue in the homology model. Briefly, chlorine positions were taken by aligning the modelled residue with a crystal structure of a halogenated tyrosine (PDB: 4NX2)[54] (Fig. 4b) and partial charges for the backbone were calculated (Supplementary Tables 5–6). Detail description of the molecular modeling protocol is found in supplementary information. These models of wild type and halogenated FtsZ monomers and dimers, were subjected to 500 ns molecular dynamics (MD) simulations (see Supplementary Methods) to gain insight into protein dynamics and GTP binding.

When aligning the halogenated and wild type monomer models after 500 ns, we did not observe drastic structural changes (Fig. 4d). On the contrary, all models were stable throughout the simulations: the root-mean square deviation (RMSD) of the monomer backbone remained below 3 Å when the disordered C-terminal tail and N-terminus were excluded (Fig. 4d–f). Halogenation may lead to more subtle changes while maintaining structural stability. To investigate this possibility, we analyzed the electrostatic potential surface (EPS) using the APBS PyMOL plugin[55]. After mutation, the surface charges of the protein mainly change around the GTP binding pocket and the N-terminal domain interface in the dimer structure (Supplementary Fig. 7).

Because GTP forms a large part of the dimer interface and GTP hydrolysis is key to monomer dissociation, we postulated that halogenation may affect GTP binding and thus cause the altered dynamics. Therefore, we calculated the solvation binding energies of the FtsZ-

GTP interactions throughout the last 100 ns of the MD trajectories (Supplementary Fig. 8, Fig. 4c). We found that both monomer and dimer models of FtsZ(Y222Cl$_2$Y) exhibited stronger GTP binding energy than the wild type. A stronger GTP-monomer association could increase the likelihood of polymerization; by lowering the free energy barrier of nucleation when two monomers come together in the presence of GTP. Furthermore, we hypothesize that the stronger association leads to tighter binding and inhibits the conformational dynamics required for GTPase activity, thereby increasing the activation energy of hydrolysis.

The dimer simulations showed overall stability, but closer examination revealed that subtle conformational changes within the monomer units were amplified when the monomers interacted with each other (Fig. 4g–j). In particular, when the N-terminal domain (residues 379–561) of the upper monomer was aligned to the energy minimized model, a significant bending of the overall wild type dimer structure was detected (Fig. 4g), while less conformational distortion is predicted for the FtsZ(Y222Cl$_2$Y) variant (Fig. 4h). This result is consistent with experimental observations that the Cl$_2$Y modification reduces the curvature of the filament and thus inhibits the ring formation (Fig. 2). Comparison of the C-terminal domain of the lower monomer with the N-terminal domain of the upper monomer revealed that the FtsZ(Y222Cl$_2$Y) monomers remained in a similar conformation and quickly reached thermodynamic equilibrium with little variation in RMSD, while RMSD in the case of wild type gradually increased from 2 Å to 5 Å and reached equilibrium at ~300 ns (Fig. 4i). This observation suggests that a slower adaptation to favorable conformations was necessary for the wild type. In addition, it agrees well with previous studies that reported the existence of two conformations of the monomer, emphasizing that a transition between the two states is important for treadmilling[41,42]. However, the slower adaptation predicted for the wild type FtsZ was not detected for the FtsZ(Y222Cl$_2$Y) dimer on the basis of RMSD evolution, indicating that conformation dynamics were inhibited by the halogenation. As consequence, the structure of the halogenated mutant appeared to be more rigid than the wild type, thereby leading to weaker bending. This can be explained by subtle conformation changes in the monomer, such as the reorganization of hydrogens bonds and other non-covalent interactions due to the altered electrostatics around the introduced modification (Supplementary Fig. 7), which further causes larger conformational differences during protein assembly (Fig. 4j).

### Discussion

In conclusion, we successfully developed a platform for site-specific incorporation of various halogenated tyrosine analogues into recombinantly expressed proteins with high fidelity and efficiency in *E. coli*. Reprogrammed translation resulted in fairly clean biosynthesis of halotyrosine-containing protein samples without the need for a rather harsh and nonspecific PTM, such as treatment of proteins with hypohalous acids. To elucidate the collective effects of halogenations, we

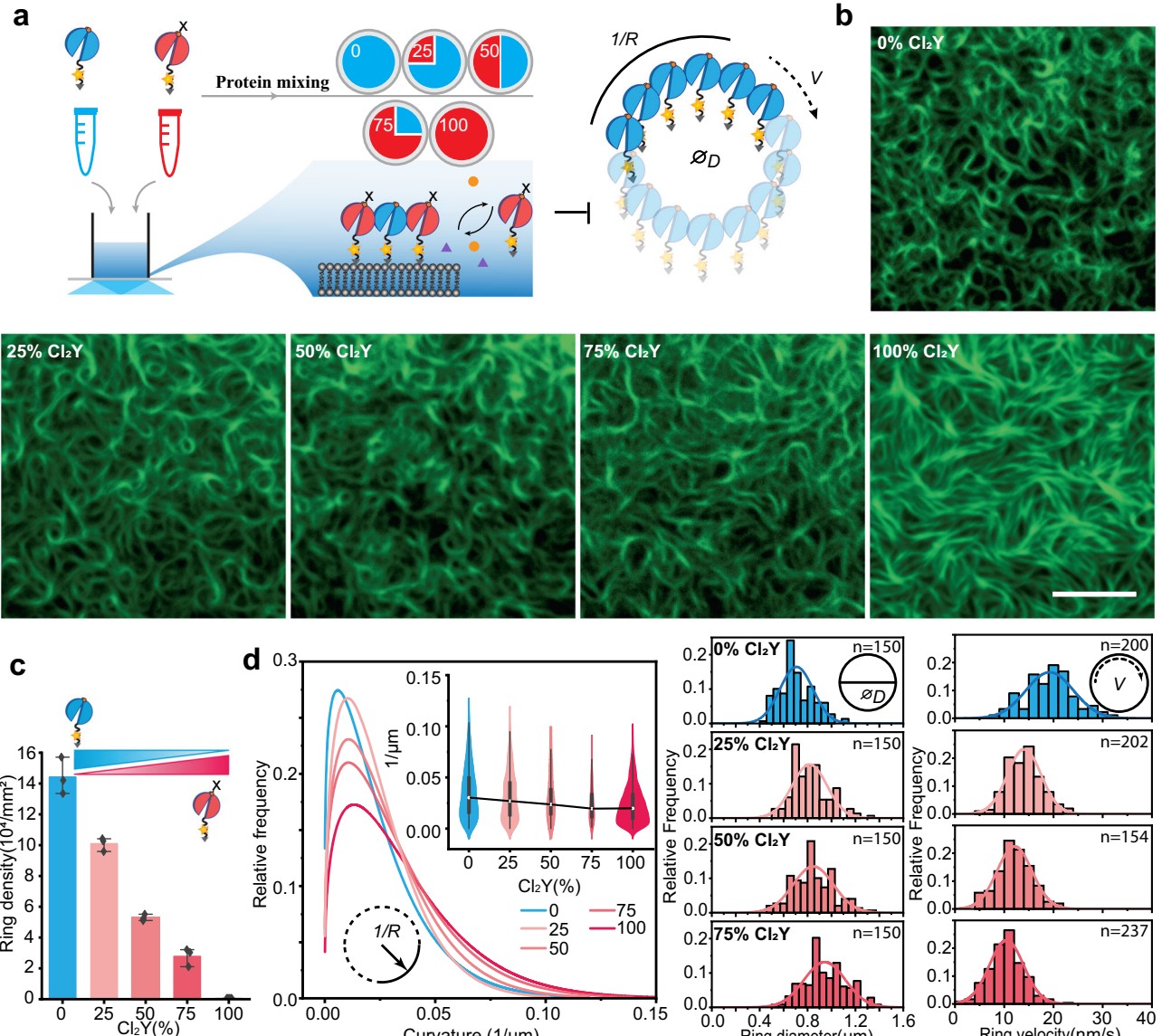

**Fig. 3 | Partial halogenated FtsZ disrupts dynamic pattern formations of wild type FtsZ. a** Schematic representation of the self-organization assay with wild type FtsZ-YFP-mts (Y) and FtsZ(Y222Cl₂Y)-YFP-mts (Cl₂Y) at certain proportions. **b** Representative cytoskeleton images of wild type FtsZ-YFP-mts and FtsZ(Y222Cl₂Y)-YFP-mts at certain proportions on supported membrane (0.5 μM proteins, 4 mM GTP and 1 mM Mg²⁺). Scale bar: 5 μm. The experiment was performed three times under identical conditions. **c** Ring densities for wild type FtsZ-YFP-mts mixed with FtsZ(Y222Cl₂Y)-YFP-mts. Data from three independent replicates are shown as means with standard deviations. **d** Distributions of curvature (left), ring size (middle), and ring velocity (right) of wild type FtsZ-YFP-mts mixed with FtsZ(Y222Cl₂Y)-YFP-mts in certain proportions. Curvatures in **d**. were calculated for the mixture of ring and filaments, representing the overall curvatures of the bulk reaction. The solid curves in **d** (left panel) represent the extreme fit for the histograms ($n_{100} = 829$, $n_{75} = 781$, $n_{50} = 312$, $n_{25} = 217$, $n_0 = 1475$). The histograms of the curvatures are not shown. Box plots in **d** (inset of left panel): the lines represent medians, box limits represent quartiles 1 and 3, whiskers represent 1.5 × interquartile range and points are outliers. The solid curves in the middle and right panels represent the Gaussian fit. Source data of **c**, **d** are provided as a Source Data file. *D*, *V* and 1/*R* in Fig. 3 indicate the ring diameter, velocity and curvature separately.

demonstrated an in vitro protein self-organization assay that can amplify the structural perturbations caused by PTMs at the atom-scale and allow us to detect the structural changes with optical microscopy. Through the in vitro protein assay and theoretical structure simulation, we discovered that even one or two newly introduced halogen atoms could readily alter the enzymatic activity of the protein, amplify or propagate their effects through protein-protein interactions, and subsequently produce global effects on protein pattern formation and dynamics. The in vitro methodology used here is not limited to a quantitative and conceptual understanding of halogenation, but could also be applied to other types of post-translational modification occurring in natural proteins. Furthermore, as a kind of fine-tuning

tools[56] our halogenated tyrosine enables the study of complex protein systems in vitro and in vivo with residue precision. The different variants of halogen atoms and modified positions will allow production of a large repertoire of new modification functions in the future. Specifically, HYs are typical products of the oxidation of the tyrosine residue in proteins, ultimately contributing to aging processes[3]. The site-specific modification of protein structures and activities in the presence of HYs should therefore enable a much better understanding of the oxidative damage-related diseases such as aging and cancer. Accumulation of this information will further help improve diagnostic methods and therapeutic interventions for patients in the future[57–59].

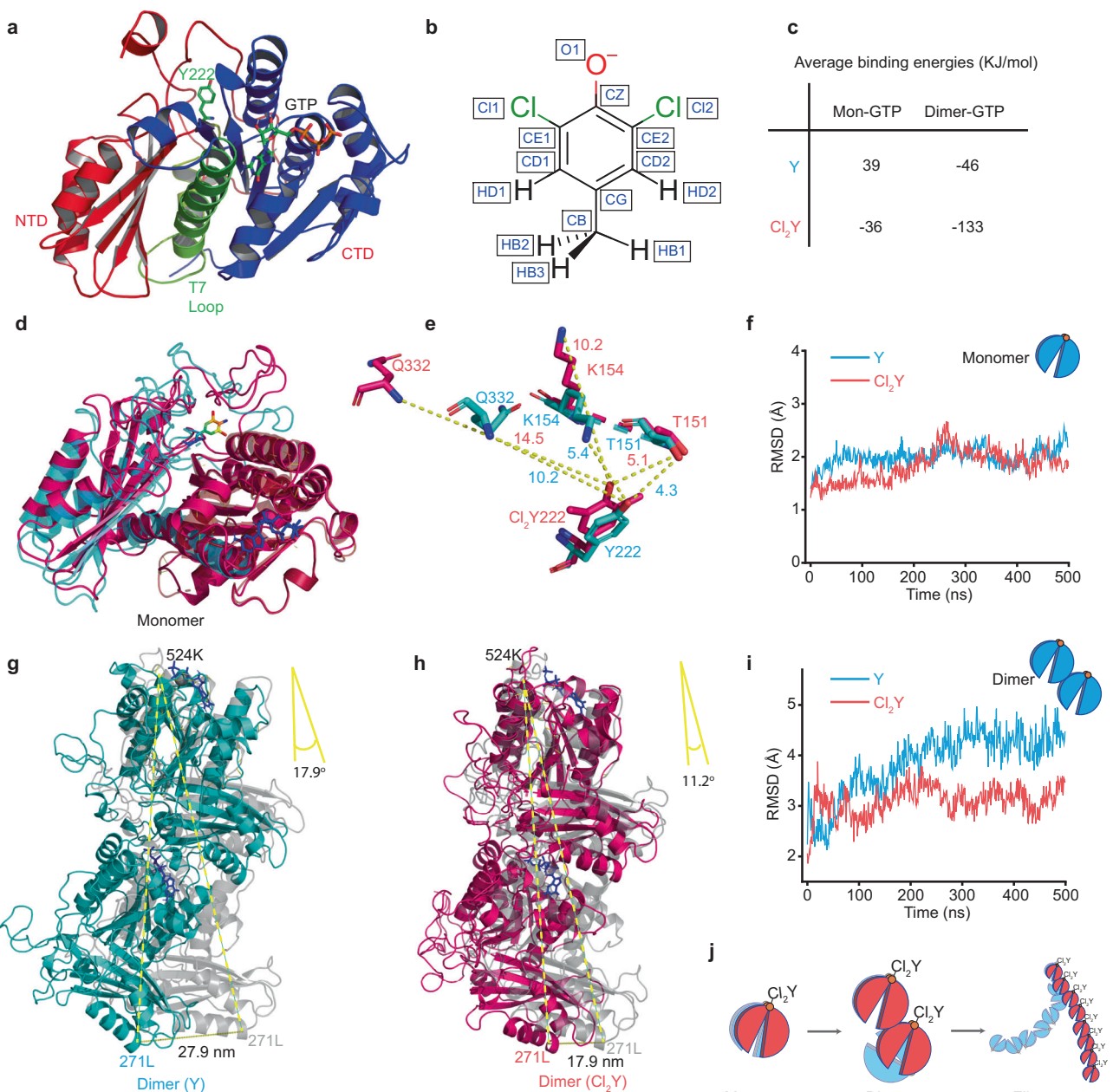

**Fig. 4 | Structure simulations of halogenated FtsZ. a** Monomer structure of wild type FtsZ. The main protein body is shown in cartoon representation, GTP and Tyr222 are shown in licorice representation. The NTD is colored red, CTD is blue and the central H7 helix is shown in green. **b** Charge calculation of 3,5-dichloro-tyrosine. **c** Average binding energies calculated according to the protocol descri-bed in Methods section using the Adaptive Poisson-Boltzmann Solver (APBS) software with time frames from the last 100 ns of MD simulations. Two sets of interactions were analyzed: Mon-GTP describes the interaction between monomer body and $Mg^{2+}$-GTP; Di-GTP illustrates the interactions between dimer body and $Mg^{2+}$-GTP at dimer interface. **d** Structural alignment of FtsZ(Y222$Cl_2$Y) (Red) with the wild type FtsZ (cyan). **e** Overlay of the mutant sites of wild type FtsZ(cyan) and FtsZ(Y222$Cl_2$Y) (red), showing the conformational changes induced by the Y222$Cl_2$Y mutation. **f** Time evolution of the backbone RMSD of FtsZ monomer (backbone atoms only) after alignment to the energy-minimized FtsZ(Y222$Cl_2$Y) model. Residues 13 to 316 are shown, excluding the disordered regions of sequence (head and C-terminal tail). **g, h** The bending motion of dimer models. Dimer alignments of **g** wild type FtsZ (cyan) and **h** FtsZ(Y222$Cl_2$Y) (red) with their energy-minimized models (grey). **i** Time evolution of the backbone RMSD of the FtsZ dimer when aligned to same selection of reference dimer structures, after energy mini-mization using AMBER software[66] in the dielectric constant ($\varepsilon = 4$). Residues 13−316 are shown for each monomer. **j** The scheme illustrates protein self-organization amplify the changes of structure by site-specific halogenation. Source data of **f** and **i** are provided as a Source Data file.

## Methods
### Materials
Primers were synthetized by Integrated DNA Technologies. Sequences of genes used in this research can be found in the sup-plementary information. All the plasmids used in this study can be found in Supplementary Table 8. Unless indicated, all standard chemicals were purchased from Sigma-Aldrich, New England BioLabs (NEB), and Thermo, Inc. Halogenated Tyr analogs (HYs) were purchased from Sigma-Aldrich, abcr GmbH, and TCI: 3-chloro-L-tyrosine (ClY, 97%, CAS 7423-93-0), 3,5-dichloro-L-tyrosine ($Cl_2$Y, ≥98.0%, CAS 15106-62-4), 3-bromo-L-tyrosine (BrY, 97%, CAS 38739-13-8), 3,5-dibromo-L-tyrosine ($Br_2$Y, ≥96.0% CAS 300-38-9), 3-iodo-L-tyrosine (IY, 95%, CAS 70-78-0), 3,5-diiodo-L-tyrosine ($I_2$Y, ≥98.0%, CAS 18835-59-1)

## One-round positive selection of *Mj*TyrRS variants

For one-round positive selection (Supplementary Fig. 1a), a dual-reporter selection plasmid pPAB26 (Supplementary Fig. 1c) was constructed, which encodes both small ubiquitin-like modifier tagged superfolder green fluorescent protein (SUMO-sfGFP) with amber codon at Arg2 position driven by T5 promoter (can elicit gene expression in DH10b) and chloramphenicol acetyltransferase reporter cassettes containing the TAG stop codon at Gln98, Asp181[60]. The experiment design implies that the synthetase-dependent suppression of amber stop codons (TAGs) should result in the cell survival and production of a fluorescence signal at the same time. The *Methanocaldococcus jannaschii*-Tyrosyl-tRNA-Synthetase (*Mj*TyrRS) was used in this study. *Mj*TyrRS gene library was constructed on pBU18'1GK plasmid (pBU18'1GK_MjTyrRS_library). One round-positive selection was performed in freshly prepared electro-competent DH10b carrying the selection plasmid pPAB26 by transformation of 100 ng *Mj*TyrRS gene library (Supplementary Fig. 1b and Supplementary Table 1)[61]. Cells were grown on new minimum medium (NMM[62]) agar plates (24 cm × 24 cm) with 1 mM Br$_2$Y or Cl$_2$Y, 100 μg/mL Ampicillin (Amp) (for propagation of the library plasmid), 50 μg/mL Kanamycin (Kan) (for maintaining positive selection plasmid), 70 μg/mL chloramphenicol (Cm) (for positive selection pressure) and 0.5 mM isopropyl β-D-1-thiogalactopyranoside (IPTG) under 37 °C for 24 h, while the control plate did not contain the Br$_2$Y or Cl$_2$Y. After selection, more fluorescent colonies were observed on the plate in the presence of Br$_2$Y or Cl$_2$Y compared to the control. Green fluorescent colonies from the Br$_2$Y or Cl$_2$Y containing plate were randomly selected and subsequently screened by streaking on NMM plates supplemented with Cm (70 μg/mL) with or without 1 mM Br$_2$Y or Cl$_2$Y. 37 out of 100 clones showed Br$_2$Y-dependent survival and green fluorescence. Meanwhile, 27 out of 66 colonies can grow in the presence of Cl$_2$Y and generate green fluorescence (Supplementary Fig. 2a). The *Mj*TyrRSs from active clones (abbreviated as BRS and CRS) were sequenced for analysis of the mutations in the synthetase. As a result, 9 unique synthetase (Supplementary Table 2) mutants were obtained after comparing the sequences of all the synthetases screened against Cl$_2$Y and Br$_2$Y. Nine selected clones containing unique synthetase genes of interest were directly applied to a 96-well fluorescence assay to test the incorporation efficiency or poly-specificity of screened *Mj*TyrRSs. The selected *Mj*TyrRSs specific for HYs were then cloned into pULTRA[63] for halogenated FtsZ production.

## Analysis of HYs incorporation through intact cell fluorescence assay

The intact cell fluorescence assay was performed to approximately compare the Br$_2$Y and Cl$_2$Y incorporation efficiency of screened *Mj*TyrRSs as well as to estimate their promiscuity against other Tyr analogues. Since the gene expression of SUMO-sfGFP was driven by T5 promoter on the one round positive selection plasmid, SUMO-sfGFP can be directly expressed in DH10b strain. Then clones containing the 9 unique *Mj*TyrRSs (pBU18'1GK plasmid) and positive selection plasmid (pPAB26_cat (Q98TAG, D181TAG) *Mj*tRNATyrCUA-his-SUMO-sfGFPR2TAG-strep), which selectively grew on the plates in the presence of Br$_2$Y or Cl$_2$Y were directly cultured in LB medium with Amp and Kan at 37 ˚C, overnight. Next day, an overnight culture was inoculated in a well at a ratio of 1:100 containing 300 μL TB medium supplemented with 50 μg/mL Kan,100 μg/mL Amp, 0.5 mM IPTG as well as 1 mM HYs while the corresponding control lacked the HYs. Cells were grown in 96 well plates (Ibidi) with orbital shaking (2 mm amplitude) for 24 h at 37 °C covered with a gas-permeable foil (Sigma Aldrich, Taufkirchen, Germany). The optical density OD$_{600}$ along with the fluorescence of the bacterial cultures were directly measured via bottom reading using excitation and emission wavelengths of 481 ± 4.5 nm and 511 ± 10 nm, respectively, and a fixed manual gain of 85. After 24 h incubation, the optimal density and fluorescence was measured via top reading after the gas-permeable foil was removed. Each measurement was done in triplicate.

## SUMO-sfGFP protein expression and purification

The clones containing the pBU18'1GK_B48RS (for ClY, Cl$_2$Y, BrY, Br$_2$Y, and IY incorporation) or pBU18'1GK_C64RS (for I$_2$Y incorporation) plasmids together with positive selection plasmid were directly cultured in LB medium with Amp and Kan at 37 °C, overnight. 1:50 or 1:100 of preculture and was transferred to TB medium. Afterwards, culture was shaken at 37 °C until OD$_{600}$ reaching around 1. Then 1 mM HYs were added directly into culture before the target gene induction. The culture was shaken continuously for another 30–40 min to allow dissolving and cellular uptake of HYs. Protein expression was induced by 1 mM IPTG for overnight at 37 °C. The SUMO-sfGFP was purified by Ni-NTA affinity chromatography (Immobilized metal ion chromatography: IMAC).

## Living *E. coli* imaging and sample preparations

Prior to imaging, all cells were grown from a single colony in LB media overnight at 37 °C. For our default slow growth condition, cells were then diluted in M9 minimal media supplemented with 0.4% Glucose, thiamine and biotin, and proper antibiotics for plasmids maintenance. Then cells were grown at 37 °C until OD$_{600}$ between 0.2–0.5. The expression of wild type FtsZ was induced with 20 μM IPTG, grown at room temperature for 3 h. For halogenated FtsZ variants expression, HYs (2 mM) were added directly into culture before the target gene induction. Then the culture was shaken continuously for another 30 min to allow dissolving and cellular uptake of the HYs. Then halogenated FtsZ was induced with 20 μM IPTG at 37 °C for 3 h. Afterwards, 1 ml cells were collected and wash twice with PBS buffer and finally resuspended in PBS. The cells were dropped on the Poly-D-lysine coated coverslips and imaged with confocal microscopy. FRAP in Supplementary Fig. 5h–i was achieved by focusing a pulsed white light laser to a diffraction-limited spot on the specimen for an exposure of ≈200 ms. Confocal fluorescent images of the cells were acquired before and after photobleaching on a Leica SP8 confocal microscope equipped with an 100x HC PL APO oil objective (NA 1.49). Images of cells were obtained every 3 s for 30 s.

## FtsZ protein expression and purification

The selected synthetase B48RS and C64RS were cloned into pULTRA plasmid. This highly efficient suppressor plasmid, pULTRA, harbors a single copy each of the tRNA and aaRS expression cassettes that exhibits higher suppression activity than its predecessors[63]. Wild type FtsZ-YFP-mts was cloned in pET-11b expression vector and transformed into *E. coli* strain BL21(DE3). Proteins were induced by 0.5 mM IPTG at 20 °C for overnight. The gene of FtsZ(Y222TAG)-YFP-mts containing an amber codon at Y222 position was cloned on pET-28a expression vector and co-transformed with pULTRA_B48RS_tRNATyrCUA (for ClY, Cl$_2$Y, BrY, Br$_2$Y, and IY incorporation) or pULTRA_C64RS_tRNATyrCUA (for I$_2$Y incorporation) into RF1 free BL21(DE3) (B95.ΔA)[64]. A preculture (1–50 mL LB with 25 mg/mL Kan, 100 mg/mL Spec and 1% glucose) was grown overnight. 1:50 or 1:100 of preculture and was transferred to TB medium. Afterwards, culture was shaken at 37 °C until OD$_{600}$ reaching around 1. Then 1 mM HYs were added directly into culture before the target gene induction. The culture was shaken continuously for another 30–40 min to allow dissolving and cellular uptake of HYs. Protein expression was induced by 0.3 mM IPTG for 3.5 h at 37 °C. Then FtsZ chimeric proteins were purified as previously described[65]. Briefly, protein was precipitated from the supernatant by adding 30% ammonium sulphate at 4 °C. Afterwards, the precipitate was shaken for 20 min at 4 °C with slow speed. After centrifugation and resuspension of the pellet, the protein was purified by anion exchange chromatography using a

5 ml Hi-Trap Q-Sepharose column (GE Healthcare, 17515601). Purity of the protein was confirmed by SDS-PAGE and mass spectrometry.

## Mass-spectrometry

Mass-spectrometry Intact mass measurements of purified proteins were performed by electrospray LC-MS on an Agilent 6530 QTOF instrument coupled with an Agilent 1260 HPLC system after external calibration. 80–100 µL of a protein solution with a concentration around 0.1 mg/mL was prepared. Samples were infused at a flow rate of 0.3 mL min$^{-1}$ onto a gradient from 5% acetonitrile 0.1% formic acid in water to 80% acetonitrile 0.1% formic acid in water through a C5 column, 2.1 × 100 mm, 3 micron (Supelco analytical, Sigma-Aldrich) over 20 minutes. The protein was ionized via electrospray ionization (ESI). Spectra deconvolution was performed with Agilent MassHunter Qualitative Analysis software (v. B.06.00, Bioconfirm Intact mass module) employing the maximum entropy deconvolution algorithm.

## GTPase activity assay of FtsZ-YFP-mts

GTPase activities of FtsZ-YFP-mts were determined by measuring released inorganic phosphate using BIOMOL® GREEN assay (Enzo Life Sciences USA). GTP hydrolysis reaction was performed in polymerization buffer (50 mM Tris/HCl, 300 mM KCl, 5 mM Mg$^{2+}$, pH7.5) using FtsZ-YFP-mts at 5 µM with 1 mM GTP. Reactions were performed every 20 s for a total time of 120 s. After 30 min of incubation with BIOMOL® GREEN at room temperature, the absorbance of the samples at 620 nm was monitored by TECAN plate reader. The phosphate release concentrations were calculated based on a phosphate standard curve.

## Dynamic light scattering (DLS)

DLS of FtsZ was measured by a Protein Solutions DynaPro MS/X instrument (Wyatt) at 25 °C using 90° light scattering cuvette. FtsZ was added in SLB buffer to a final concentration of 12.5µM, and samples were measured using a fluorometer cuvette with a 1-cm path length. Prior to measurements, samples were centrifuged for 10 min at $10^5 \times g$. Excitation and emission wavelengths were set to 350 nm, with slit widths of 5 nm. Data was collected for 5 min to get baseline. Afterwards, 4 mM GTP was added to achieve a final reaction volume of 300 µl. The reaction mixture was gently stirred and returned to chamber for data collection for at least 70 min.

## Small unilamellar vesicles (SUVs) preparation

1,2-dioleoyl-sn-glycero-3-phosphocholine (DOPC):1,2-dioleoyl-sn-glycero-3-phospho-(1′-rac-glycerol) (DOPG), 70:30 mol % mixture, was dissolved in chloroform in a quartz container, then dehumidified under a gas nitrogen stream. Chloroform traces were further removed through desiccation (1 h). Afterwards, the lipid film was hydrated to a final concentration of 4 mg/ml in supported lipid bilayers (SLB) buffer (50 mM Tris-HCl at pH 7.5, 150 mM KCl), and incubated at 37 °C for 30 min. The lipid film was then completely resuspended by vortexing rigorously to obtain multilamellar vesicles of different sizes. This mixture was then placed in a bath sonicator where shear forces help to reduce the size of the vesicles, giving rise to small unilamellar vesicles (SUVs). The SUV dispersion were stored at −20 °C as 20 µl aliquots.

## Supported lipid bilayer (SLB) preparation

Glass coverslips (1.5#, 24 x 24 mm) were cleaned by piranha solution overnight, followed by extensive washing with milliQ H$_2$O. Then glass coverslips were blown dry with compressed air. The reaction chamber was prepared by attaching a plastic ring on a cleaned glass coverslip using ultraviolet glue (Thorlabs No. 68). For supported lipid bilayer formation, the SUV dispersion was diluted in SLB buffer to 0.5 mg/ml, of which 75 µl was added to the reaction chamber. Adding CaCl$_2$ to a final concentration of 3 mM induced fusion of the vesicles and the

formation of a lipid bilayer on coverslide. After 20 min of incubation at 37 °C, the sample was rinsed with 2 ml pre-warmed SLB buffer.

## Self-organization assays

FtsZ-YFP-mts was added to the reaction buffer above the supported lipid membrane in the chamber. The final volume of a sample was approximately 250 µl. FtsZ-YFP-mts was added with a final concentration of 0.5 µM. Polymerization was induced by adding 4 mM GTP.

## Total internal reflection fluorescence microscope (TIRFM) imaging

All experiments were performed on a WF1 GE DeltaVision Elite Total internal reflection fluorescence microscope (TIRFM, GE Healthcare Life Sciences, Germany) equipped with an OLYMPUS 100× TIRF objective (NA 1.49). The UltimateFocus feature of DeltaVision Elite maintains the focus plane constant in time. FtsZ-YFP-mts was excited with a 488 nm diode laser (10 mW before objective). Fluorescence imaging was performed using a standard FITC filter set. Images were acquired with a PCO sCMOS 5.5 camera (PCO, Germany) controlled by the softWoRx Software (GE Healthcare Life Sciences, Germany). For time-lapse experiments, images were acquired every 3 s with a 0.05 s exposure time and light illumination shuttered between acquisitions.

## Ring velocity analysis and processing

FtsZ ring velocities were analyzed according to the published approach[47]. Briefly, image analysis was carried out in MATLAB 2016s (MATLAB and Image Processing and Computer Vision Toolbox Release 2016a, The MathWorks, Inc., Natick, Massachusetts, USA) and processing with Fiji/ImageJ(1.53f51). Images corresponded to an average of 5–10 frames from a time-series experiment. For the kymograph analysis, time-series acquisitions were filtered using a standard mean filter (2 pixel) and were drift corrected (multistackreg plugin). A MATLAB script allowed the user to define a ring by providing two coordinates. Every ring was automatically fitted to a circle with radius r. Then, three trajectories corresponding to three concentric circles having radii r, r + 1, and r − 1 pixels were determined. At this point, the script read the time-series data and calculated a kymograph for each time point and trajectory. To automatically calculate the slope, the kymograph was smoothed with a Savitzky-Golay filter of order 2 and enhance its contrast using a contrast-limited-adaptive-histogram-equalization (CLAHE) routine (MATLAB). Next, using Fourier analysis from previous study[47], the characteristic frequency for the patterns on the kymograph was found. Finally, the slope corresponded to the change in phase at this frequency. Quality criteria were properly chosen to reject low-quality regions over the kymograph. To synchronize time-lapse acquisitions, the initial frame (time 0) was defined when surface mean intensity was around 200 A.U.

## Fluorescence recovery after photobleaching data (FRAP)

Fluorescence recovery after photobleaching data (FRAP) on the SLB were evaluated by choosing two separately circle areas ($r = 4$ µm). One circle was taken as a reference. Another one was photobleached by 20 iterations of 488 nm, laser under 100% laser power. Their fluorescence recovery in the green channel was monitored. Intensity traces were collected using Fiji, corrected for photobleaching and normalized with the pre-bleaching intensity and the reference intensity. The fluorescence recovery curves were then fitted with mono-exponential equation using Originpro (v_2017 & v_2019):

$$Y = Ae^{-kt} + B, \tag{1}$$

where $Y$ represents fluorescence intensity at time $t$, $A$ and $B$ are parameters, and $k$ represents the rate constant. The half-life, $t_{1/2}$, was

determined using the following equation:

$$t_{1/2} = ln2k^{-1} \qquad (2)$$

## Side-chain acidity

10–20 mg of an amino acid, 50 mg of $KH_2PO_4$ was dissolved in 15 ml of water and titrated by sodium hydroxide solution to distinct pH values. 500 µl were taken to NMR tubes, 50 µl deuterium oxidewas added and $^1H$ NMR spectra were measured using W5 pulse tray water suppression at 500 MHz frequency. The chemical shifts were plotted against pH and analyzed using Henderson-Hasselbalch equation to deliver the $pK_a$ values. Experimental error ±0.15.

Ammonium $pK_a$: Y 9.44, ClY 9.50, $Cl_2$Y 9.40, BrY 9.52, $Br_2$Y 9.47, IY 9.37, $I_2$Y 9.38.

Phenolic $pK_a$: Y 9.90, ClY 8.31, $Cl_2$Y 6.52, BrY 8.29, $Br_2$Y 6.42, IY 8.30, $I_2$Y 6.45.

## Lipophilicity analysis

The lipophilicity of the amino acid residues was measured against 150 mM buffers as described[32]. More details can be found in the supplement methods. N-Acetyl-tyrosine methyl ester and N-acetyl-3-iodo-tyrosine methyl ester were reported earlier, and other compounds were synthesized according to the protocol. The distribution coefficients are provided in Supplementary Table 7.

## Statistics and reproducibility

Unless otherwise mentioned, all measurements were performed for at least three independent experiments. The numbers of experimental repeats are indicated in the respective figure legends. The statistical tests were analyzed by ANOVA one-way statistical tests (significance level, 0.05; mean comparison, Tukey; tests for equal variance, Levene; power analysis, actual power). All $p$ values are given in the respective figure captions.

## Reporting summary

Further information on research design is available in the Nature Research Reporting Summary linked to this article.

## Data availability

All data used in this paper are available at Figshare through the identifier [https://doi.org/10.6084/m9.figshare.20368626.v1] or from the corresponding authors upon request. Source data are provided with this paper. All the structures used in this study are publicly available under the PDB accession codes 1W5A (FtsZ dimer, MgGTP soak), 4NX2 (Crystal structure of DCYRS complexed with DCY) and 3GFD (Crystal structure of IYD bound to FMN and mono-iodotyrosine MIT) Source data are provided with this paper.

## Code availability

No custom code was used in this research. The previously published code of ring velocity analysis is available upon request from author (Prof. Petra Schwille, schwille@biochem.mpg.de, PLoS biology, 2018, 16(5): e2004845).

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

## Acknowledgements

Huan Sun was supported by China Scholarship Council. Haiyang Jia was supported by the GRK2062 Molecular Principles of Synthetic Biology, funded by Deutsche Forschungsgemeinschaft (DFG). This work is also a part of the MaxSynBio consortium which is jointly funded by the Federal Ministry of Education and Research of Germany and the Max Planck Society. Nediljko Budisa and Vladimir Kubyshkin thank Canada Research Chairs Program (Grant No. 950-231971) for support. Jovan Dragelj and Andrea Mroginski thank the Deutsche Forschungsgemeinschaft (DFG) - EXC 2008-390540038-UniSysCat for financial support. We thank Dr. Diego A. Ramirez-Diaz for providing the ring velocity analysis code.

## Author contributions

Conceptualization, H.J., H.S., P.S., M.A.M. and N.B.; Investigation—Experiment, H.J., H.S. and V.K.; Investigation—Simulation, J.D., and O.K.; Methodology, H.J., H.S., J.D. and T.B.; Project Administration, H.J., P.S., M.A.M. and N.B.; Funding and Resources, P.S., M.A.M. and N.B.; Visualization, H.J. and H.S.; Writing—Original Draft, H.J. and H.S.; Writing—Review & Editing, H.J., H.S., V.K., J.D., O.K., M.A.M., P.S. and N.B.

## Competing interests

The authors declare no competing interests.
