## [Peer Review File · Nature Communications]

REVIEWER COMMENTS

Reviewer #1 (Remarks to the Author):

Many kinds of proteins assemble into higher-order architectures in living cells. Post-translational modifications (PTMs) can affect protein structures and their interactions with other molecules, thus regulating cell functions or causing diseases in particular cases. Genetic code expansion makes it possible to generate PTMs on premeditated sites in proteins and investigate their specific effects. Although there have been many studies to explore such effects on individual proteins, what influences PTMs can exert on higher-order cellular structures were rarely studied. Since some protein modifications are implicated in degraded cell functions and possibly aging as a result, the research direction pointed in the present study is relevant to modern issues.

The authors took cell division machinery to obtain insights into the influences of tyrosine halogenation (an age- and disease-related PTM) on protein self-assembly. First, the authors engineered novel aminoacyl-tRNA synthetases to incorporate a series of halogenated tyrosines into self-organizing FtsZ proteins. Some of these tyrosine derivatives were thus included in expanded genetic codes for the first time. Using this tool, they analyzed the effects of halogenation at a particular site of FtsZ on its GTPase activity, in-vivo and in-vitro assembly of division ring, and finally on structural dynamics in computer simulations. Thus, they showed that a tiny posttranslational alteration (the addition of just one or two halogen atoms to an entire protein molecule) can have a profound effect on the formation of cellular apparatus. A lot of sound data are presented to support conclusions. The computer simulations were particularly helpful for understanding how subtle changes at the atomic level were amplified in the process of protein assembly. I believe that the manuscript is worth publishing after the authors addressed the points below.

Here are a few suggestions that might improve the manuscript.

1) Expanded genetic codes have been employed to elucidate the natural or disease-related roles of PTMs. Lysine acetylation and methylation have been generated site-specifically on histones to analyze their influences on nucleosome assembly and interaction with histone modification factors. Nitrotyrosine has been generated in superoxide dismutase from mitochondria and shown to thwart its enzymatic activity (DOI: 10.1021/ja710100d). The authors should mention these previous achievements, probably in the introduction.

2) The main point of this study is about the effect of a subtle change in protein structure amplified at the higher level of molecular assembly. This can also occur with mutations causing amino acid replacements.

I would like to ask the authors to discuss what is special about PTM in such effects. How are PTMs different from mutational amino acid replacements in causing higher-order effects?

3) In Figure S6, Y222 and GTP should be indicated in each representation for the reader's convenience.

4) The label "a" is wrongly put on the structure in the middle of the 3rd row of this figure. And the Cl2Y-mutant FtsZ strangely looks much bigger than the WT molecule in the "a" row.

5) In the "c" row of Fig. S6, the top right part of the Cl2Y-FtsZ structure looks very different from the WT counterpart. Could this be relevant to the effect of halogenation on the GTPase activity and FtsZ self-assembly?

6) Fig. 4d also shows a similar structural change around the top between these two FtsZ molecules. Could this change be relevant to the weaker bending of the Cl2Y mutant shown in Fig. 4h?

7) The cause of this weaker bending in the Cl2Y mutant dimer is not discussed in the text. The authors wrote, "Y222Cl2Y-FtsZ displayed little bending and thus its structure appeared more rigid than the wild-type (l. 306)". If this structural rigidity is the cause of the weakly bend dimer structure, how did the halogenation bring about that rigidity?

8) At first, I did not understand why Tyr222 had been chosen for the modification. Probably, this Tyr is the only Tyr near the domain-domain interface suitable for replacements with halogenated derivatives. This should be mentioned in the text. Tyr222 seems to be exposed to the solvent, judging from the presented structures. This should be mentioned too, because, otherwise, this position could have no access to external natural modifiers.

9) I did not understand the term "collective" in the title before reading the text. The authors could reword it.

10) Regarding the terms "essential tyrosine" in the title and "at a key position" in the abstract and main text, what makes Tyr222 essential and position 222 a key position? Tyr222 is nothing more than the only Tyr located at the domain interface.

11) The abstract does not mention FtsZ in it at all. So, the reader can have no idea what the details (structural bending, GTPase activity, etc) in the abstract are all about.

12) “filamenting temperature-sensitive mutant Z” is the name of a mutant. FtsZ is mentioned as a particular protein (the protein product of the ftsZ gene) throughout the text. I suggest that lines 70-71 be changed to “...based on the FtsZ protein (the ftsZ gene product), the known prokaryotic homologue of the eukaryotic protein tubulin (Fig. 1a).”

13) “the bacterial division (Z) ring (l.73) should be changed to “the bacterial division ring (Z ring)”.

14) I do not see how the development of halogenated biologics, mentioned in the last part of the discussion, is related to the scope of this study. The authors could discuss this issue at another opportunity.

Reviewer #2 (Remarks to the Author):

Sun et al. study the effects of tyrosine halogenation of FtsZ on protein self-organisation. For this, experimental and theoretical approaches are combined. The authors state in their conclusion that even a single introduced halogen atom can alter the activity of a protein. The presented results enhance our understanding of oxidative damage-related diseases and enable future applications such as antibody engineering.

While I find the outcome of this study interesting for the general readership of Nature communications, I think the manuscript should be improved so that the non-expert reader can follow the rationale and conclusions drawn from the experiments. Please find some general and specific comments below:

- The results and discussion are described in one large section. I suggest to provide subheaders to separate the individual experiments and to structure the findings. This will help the reader to follow the results/conclusions.
- It is difficult for the non-expert reader to understand the principle of the reconstituted minimal cell division system. More explanations would be helpful.
- The latter point is true for most of the described experiments. For instance, it is difficult to understand how results presented in Figure 1c were obtained. Only the figure legend provides information on the experiment that was performed.
- The authors use different abbreviations for the halogenated tyrosine residues (e.g. ClY, Cl2Y etc). Abbreviations for the wild-type are mixed throughout the manuscript (Y versus WT versus wild-type etc). I suggest to use the same term for the non-modified protein.
- Schematics of experiments are provided (e.g. Figs 2a and 3a). I suggest to include a short explanation to the text, too.
- How are the mass differences of halogenated tyrosine variants explained? The mass difference is too high. For instance, the mass of chlorine is 35, attachment to tyrosine includes the loss of a H-atom and the observed mass difference should be 34 Da.
- Which mass spectrometer was used? The methods sections states that a Q-ToF instrument from Agilent was used, but the proteins were introduced into a mass spectrometer from Thermo Scientific. The analysis was performed with an Orbitrap but a Q-ToF was used. It is unclear how the experiments were performed. Mass spectra shown in Fig. 1d do not show the expected high resolution of an Orbitrap mass analyser. In addition, experimental details are missing (e.g. specification of the column used (particle size etc), parameters of the MS analysis (in particular ion mode)). The MS section needs to be corrected.

Reviewer #3 (Remarks to the Author):

In their manuscript Sun et al. describe the effect of tyrosine halogenation on FtsZ polymerization. This is a very interesting system to study, because FtsZ is essential in prokaryotic cell division and acts as interaction hub to recruit other cell division proteins. The authors show that mutation (halogenation) of a single residues in FtsZ causes significant phenotypic changes. The use of genetically encoded halogenated amino acids is a nice demonstration of this powerful technique to produce homogeneous samples and cleanly analyze their function. The self-organization assays are well executed and are informative. Basic molecular dynamics simulations are presented, but the complexity of the system make the results hard to interpret.

My main criticism with the manuscript is that it mainly presents observations but falls, in its current form, short in clearly rationalizing and explaining results. The authors acknowledge this themselves by stating “Because the physiological complexity it is not yet possible to derive a particular underlying mechanism from these cellular results. However, the results are sufficient to ascertain dramatic effects resulting from single site halogenation”.

Overall I believe the manuscript could be improved by the authors addressing the following points and revising the manuscript accordingly:

1. on page 5 it is stated “Thus we infer that tyrosine halogenation may dramatically affect sidechain acidity, volume and hydrophobicity, while these perturbations subsequently lead to suppressed GTPase activities.” The authors should correlate physicochemical properties of the noncanonical amino acids (Figure 1c) with their measured rates of GTP hydrolysis and rates of polymerization (Figure 1f). Neither of these properties seem to correlate with rate of hydrolysis or polymerization.
2. The selection of tRNA synthetase enzymes for the noncanonical amino acids seems unusual, because only one round of positive selection was used. Previous studies using variant libraries of the *M. jannaschii* tRNA synthetase report that the gene pool after the first positive selection round mainly contains RS enzymes that recognize canonical amino acids. A more detailed description should be included, because there is clearly important information missing. The authors should clearly state how many colonies were screened and found selective for noncanonical amino acids. Further, it should be clearly stated which plasmids are used for different experiments. I.e. “genes of interest were cloned into pULTRA for further experiments and/or directly applied to a 96-well fluorescence assay”. Is Figure 2b produced with pULTRA or pBU16 plasmids?
3. The quality of the mass spectrometry data is on the lower end of what can be expected from the specified setup. While the peak maxima correspond well with the expected masses, the mass signals are broad and not resolved. This may be a setting in the deconvolution program. Figure 1d clearly shows satellite peaks for Cl2Y and I2Y indicating dehalogenation and there are higher mass peaks consistent in all samples. Better mass spectra quality likely reveal their identity, which may also be important in the discussion of effects on function. The mass spectra in Figure S3 would be more useful if only the mass range of 40000 to 41000 Da was shown in high resolution.
4. The addition of homology modelling and molecular dynamics simulations appears to be an add-on. Details on how the MD simulations were performed are largely missing and cannot be reproduced from the current description. Molecular dynamics simulations are difficult to interpret at the best of times, and I would suggest not to rely on single trajectories for interpretation.
5. Throughout the manuscript the authors should check the precision with which experimental results are reported, and only report significant figures. I.e. In Figure 4c, are the binding energies really accurate to 0.01 kJ/mol? Provide error estimates. Observed protein masses are reported with 2 significant digits.

Reviewer #4 (Remarks to the Author):

Sun et al. presented the experimental platform along with the evidence that halogenation of a particular tyrosine residue in FtsZ at a molecular-level perturbs the self-organization of FtsZ polymers at a cellular-level. The study combined *in vivo* live-cell imaging, *in vitro* reconstitution, and molecular dynamics simulations to substantiate their finding and conclusions. Since halogenation is an important form of non-enzymatic post-translational modifications with the latter being an emerging factor in several oxidative stress-related diseases, biophysical and biochemical characterizations of the halogenation effects in this study could be of potential importance and interest in the general field of biophysics and synthetic biology. That being said, I'd be delighted to see this work in publication. However, I'd suggest the authors to consider the following suggestions to improve their work/manuscript before publication.

1. The authors elucidated the halogenation effects on the formation and GTPase-mediated treadmilling dynamics of FtsZ ring with the *in vitro* reconstitution. The biophysical and biochemical characterizations seem to be solid. But if I understand correctly, I failed to see how this particular effect on FtsZ has anything to do with the general principle of protein halogenation. Is the halogenation effect just another post-translational modification or unique in some aspect(s)?

2. In light of FtsZ, the central organizer of bacterial divisome, the authors demonstrated that different kinds of halogenations at Y222 can change the geometry of FtsZ ring and slow the treadmilling speed. I have several questions as follows.

2.1) The treadmilling speed of the wt FtsZ-YFP-MTS is ~ 20 nm/s, which is much slower than those *in vivo* (~ 30 - 40 nm/s). What causes this slower speed? Does it relate to the fact that FtsZ is linked to membrane by MTS (membrane target sequence), rather than ZipA and FtsA as that *in vivo*? If so, then how will the halogenation effect play out *in vivo* with the presence of ZipA and FtsA? Is it possible that the halogenation effects are masked by ZipA and FtsA so that treadmilling speed may not be perturbed? The authors need to demonstrate the relevance of halogenation effects on FtsZ treadmilling speed *in vivo*.

2.2) It is difficult to discern the FtsZ ring structures in Cl2Y and IY (figs 2b and 3b); instead, they look to me more like a cluster without the close loop. Why did author term these structures as rings? How did the authors quantify the ring diameter? Will it be more meaningful to quantify the length of FtsZ filament length in these clusters?

2.3) If the FtsZ ring is ~ 0.8 - 1 micron in diameter, shouldn't the curvature of the ring be $\sim 1/0.4 \text{ -- } 1/0.5$ (micron^{-1}), which is $\sim 2 - 2.5 \text{ micron}^{-1}$? However, the figs. 2f and 3d show that the curvature, if I understand it correctly, is on the order of 0.01 micron^{-1} .

3. The authors leveraged molecular dynamics simulation (by AMBER) to study how halogenations affect the energy and dynamics of FtsZ dimerization. It is intuitive that if the FtsZ dimer has a sharper kink, then a smaller FtsZ ring tends to form (aka with a higher curvature). I have two questions.

3.1) The simulation run is ~ 100 nanoseconds, and yet, the FtsZ ring formation and treadmilling is on the timescale of seconds to minutes. There is a gap between the MD simulation result and the experiment observation on cellular scales; this always leaves the interpretation of MD simulation result in some limbo. Can the authors elaborate what are expected to hold from their MD simulation result on the cellular scales? And what are not?

3.2) Why did the authors choose the dielectric constant to be 4? To put this in context, the dielectric constant is 80 for water and 2-3 for lipid bilayer/membrane. Given FtsZ polymer is periphery to membrane and hence in cytosol, the interactions within and between the individual FtsZ dimer are expected to be in water. Setting the dielectric constant to be 4, did the authors imply that the FtsZ dimer in membrane? Or is this dielectric constant referred as the internal dielectric constant within the protein? If so, then what is the physical underpinning of choosing this internal dielectric constant to be 4? More importantly, how does the simulation results in figs. 4c, f, and i change with the value of this dielectric constant?

REVIEWER COMMENTS

Reviewer #1 (Remarks to the Author):

Many kinds of proteins assemble into higher-order architectures in living cells. Post-translational modifications (PTMs) can affect protein structures and their interactions with other molecules, thus regulating cell functions or causing diseases in particular cases. Genetic code expansion makes it possible to generate PTMs on premeditated sites in proteins and investigate their specific effects. Although there have been many studies to explore such effects on individual proteins, what influences PTMs can exert on higher-order cellular structures were rarely studied. Since some protein modifications are implicated in degraded cell functions and possibly aging as a result, the research direction pointed in the present study is relevant to modern issues.

The authors took cell division machinery to obtain insights into the influences of tyrosine halogenation (an age- and disease-related PTM) on protein self-assembly. First, the authors engineered novel aminoacyl-tRNA synthetases to incorporate a series of halogenated tyrosines into self-organizing FtsZ proteins. Some of these tyrosine derivatives were thus included in expanded genetic codes for the first time. Using this tool, they analyzed the effects of halogenation at a particular site of FtsZ on its GTPase activity, in-vivo and in-vitro assembly of division ring, and finally on structural dynamics in computer simulations. Thus, they showed that a tiny posttranslational alteration (the addition of just one or two halogen atoms to an entire protein molecule) can have a profound effect on the formation of cellular apparatus. A lot of sound data are presented to support conclusions. The computer simulations were particularly helpful for understanding how subtle changes at the atomic level were amplified in the process of protein assembly. I believe that the manuscript is worth publishing after the authors addressed the points below.

We thank the reviewer for the positive feedback and insightful comments.

Here are a few suggestions that might improve the manuscript.

1. Expanded genetic codes have been employed to elucidate the natural or disease-related roles of PTMs. Lysine acetylation and methylation have been generated site-specifically on histones to analyze their influences on nucleosome assembly and interaction with histone modification factors. Nitrotyrosine has been generated in superoxide dismutase from mitochondria and shown to thwart its enzymatic activity (DOI: 10.1021/ja710100d). The authors should mention these previous achievements, probably in the introduction.

We thank the reviewer for his/her valuable suggestions. We have now added additional information in the introduction and cited the related references mentioned by the reviewer (Page 2, Line 28-Page 3, Line 5).

Page 2, Line 28-Page 3, Line 5: Co-translational modification of target proteins with oxidized ncAAs at defined positions has already proven to be a useful tool to study the role of protein nitration¹⁵ or oxidation¹⁶ at specific positions. For example, nitrotyrosine has been genetically incorporated in superoxide dismutase from mitochondria to encode the protein oxidative damage¹⁷. The effect of protein modifications in collective intramolecular processes¹⁸ in protein complexes assembly^{19,20} and in cells and tissues^{15,21,22} have also been

studied. For instance, modifications such as acetylation²⁰ and methylation¹⁹ have been used to determine the effects of modifications on nucleosome complexes assembly and cellular transcriptional responses.

2. The main point of this study is about the effect of a subtle change in protein structure amplified at the higher level of molecular assembly. This can also occur with mutations causing amino acid replacements. I would like to ask the authors to discuss what is special about PTM in such effects. How are PTMs different from mutational amino acid replacements in causing higher-order effects?

Thank you very much for this insightful comment. First, we must point out that our main purpose is to elucidate the effects of halogenation on the large-scale progress of protein assemblies. Subtle structural change is one of the effects caused by halogenation. We agree that the subtle changes of protein structures can be achieved by conventional amino acid mutations; however, the traditional mutations methods cannot give us insight for the oxidative modifications, nor can provide fine regulation of protein structures as precise as PTMs. Below, we list some of the strengths of PTMs compared with traditional mutation with natural amino acids:

1. **Minimal changes and high precision.** PTMs mainly take place on the side chains of amino acids and slightly change the structures and properties of amino acids. Compared to the mutation approach with completely different amino acids, the genetic code expansion technique offers the minimal intervention in proteins with atomic precision. Therefore, when regulating protein functions, PTMs are much more precise tools for fine-tuning tools. Genetic code expansion also allows us to unambiguously study the effects of modification with a single chemical group or atom on the side chain compared to the native form of the amino acid, e.g. by comparing Cl₂Y and Y.
2. **Diverse options and customized modifications.** The advancement of genetic code expansion now allows us to synthesize and insert amino acids that are individually modified (or custom-modified). Diverse ncAAs (>100) with versatile chemical properties have been site-specifically incorporated into protein of interest. For example, the halogenated tyrosine analogues (HYs) in this study have various halogen atom substitutions at different sites of aromatic ring. These halogenations alter the properties of both aromatic ring and hydroxyl function (pKa). Such changes in properties affect local environments in protein structure but could also be propagated to affect higher-order architectures in living cells. Therefore, these subtle atomic changes (in literature also called "atomic mutations" or "molecular surgery") are on one hand very useful non-invasive tools and on the other hand also allow the emergence of a large repertoire of new functions.
3. **Probing the role of natural post-translational modifications** in protein structure and function. Site-specific incorporation of ncAAs can provide insight into how the position, density, and distribution of protein modifications affect protein function.

We have now discussed these reasons in both the Introduction and Discussion section (Page 2, Line 26-28, Page 16, Line 18-25).

Page 2, Line 26-28: *By mimicking PTMs, the site-specific incorporation of the ncAAs can provide information on how the position, density, and distribution of protein modifications perturb protein structures and functions on a small scale.*

Page 16, Line 18-25: *Furthermore, as a kind of fine-tuning tools⁵⁶ our halogenated tyrosine enables the study of complex protein systems in vitro and in vivo with residue precision. The different variants of halogen atoms and modified positions will allow production of a large repertoire of new modification functions in the future. Specifically, HYs are typical products of the oxidation of the tyrosine residue in proteins, ultimately contributing to aging processes³. The site-specific modification of protein structures and activities in the presence of HYs should therefore enable a much better understanding of the oxidative damage-related diseases such as aging and cancer.*

3. In Figure S6, Y222 and GTP should be indicated in each representation for the reader's convenience.

We understand the reviewer's concern and agree that the labelling of important residues should be clearly indicated. We have now added the suggested correction to Fig. S6, now named as Fig. S7.

4. In Figure S6 The label "a" is wrongly put on the structure in the middle of the 3rd row of this figure. And the Cl₂Y-mutant FtsZ strangely looks much bigger than the WT molecule in the "a" row.

We have removed the wrong label and recreated the new Fig. S6, now named as Fig. S7.

Fig. S7

5. In the "c" row of Fig. S6, the top right part of the Cl₂Y-FtsZ structure looks very different from the WT counterpart. Could this be relevant to the effect of halogenation on the GTPase activity and FtsZ self-assembly?

In Fig. S6c (now Fig. S7c), the region on top right part of the FtsZ(Y222Cl₂Y) structure corresponds to loops that are flexible in MD simulations. Therefore, the structural differences predicted in this region have to be interpreted with caution, especially when comparing single conformations (as shown here). However, the residues of the loop region in question are in direct contact with the mutation site (Y222Cl₂Y) and thereby their role in the bending and conformational dynamics of the whole monomer may be relevant. While in our work we suspect that halogenation affects the GTPase activity and self-assembly, we cannot claim that this particular conformational change of the loop region is directly responsible for these effects.

6. Fig. 4d also shows a similar structural change around the top between these two FtsZ molecules. Could this change be relevant to the weaker bending of the Cl₂Y mutant shown in Fig. 4h? The cause of this weaker bending in the Cl₂Y mutant dimer is not discussed in the text. The authors wrote, “Y222Cl₂Y-FtsZ displayed little bending and thus its structure appeared more rigid than the wild-type (l. 306)”. If this structural rigidity is the cause of the weakly bend dimer structure, how did the halogenation bring about that rigidity?

As an extension of the response to comment 5, the structural changes observed at the top region of the two FtsZ molecules are common conformational changes expected within loop regions during MD simulations. However, we cannot claim that these changes are the cause of a weaker bending of the dimer as they are not located at the interface between the monomers (but ~20Å away). On the other hand, this loop region may be involved in the conformational dynamics and bending properties monomer since it is located in the region between CTD and NTD. Although the RMSD values of the backbone atoms of the Cl₂Y mutant indicate weaker bending, structural details and mechanism are not known and were not the focus of this work. Furthermore, the complete sampling of the conformational space of the loop region and the interpretation of these results would be a separated work.

In the revised version of the manuscript, we have extended the discussion on the causes for weaker bending of the FtsZ variant structure (Page 15, line 16-Page 16, line 4). As seen in our dynamic models (Fig. 4i), the introduced halogenated tyrosine mutation may affect the conformational dynamics of the FtsZ dimer structure. This can be due to subtle conformation changes in the monomer such as the reorganisation of hydrogens bonds and other non-covalent interactions due to the altered electrostatics around the introduced modification (Fig. S7), which in turn causes larger conformational differences in the dimer.

Page 15, line 16-Page 16, line 4: *In particular, when the N-terminal domain (residues 379-561) of the upper monomer was aligned to the energy minimized model, a significant bending of the overall wild type dimer structure was detected (Fig. 4g), while less conformational distortion is predicted for the FtsZ(Y222Cl₂Y) variant (Fig. 4h). This result is consistent with experimental observations that the Cl₂Y modification reduces the curvature of the filament and thus inhibits the ring formation (Fig. 2). Comparison of the C-terminal domain of the lower monomer with the N-terminal domain of the upper monomer revealed that the FtsZ(Y222Cl₂Y) monomers remained in a similar conformation and quickly reached thermodynamic equilibrium with little variation in RMSD, while RMSD in the case of wild type gradually increased from 2 Å to 5 Å and reached equilibrium at ~300 ns (Fig. 4i). This observation suggests that a slower adaptation to favorable conformations was necessary for the wild type. In addition, it agrees well with previous studies that*

reported the existence of two conformations of the monomer, emphasizing that a transition between the two states is important for treadmilling^{41,42}. However, the slower adaptation predicted for the wild type FtsZ was not detected for the FtsZ(Y222Cl₂Y) dimer on the basis of RMSD evolution, indicating that conformation dynamics were inhibited by the halogenation. As consequence, the structure of the halogenated mutant appeared to be more rigid than the wild type, thereby leading to weaker bending. This can be explained by subtle conformation changes in the monomer, such as the reorganization of hydrogens bonds and other non-covalent interactions due to the altered electrostatics around the introduced modification (Fig. S7), which further causes larger conformational differences during protein assembly (Fig. 4j).

7. At first, I did not understand why Tyr222 had been chosen for the modification. Probably, this Tyr is the only Tyr near the domain-domain interface suitable for replacements with halogenated derivatives. This should be mentioned in the text. Tyr222 seems to be exposed to the solvent, judging from the presented structures. This should be mentioned too, because, otherwise, this position could have no access to external natural modifiers.

We thank reviewer for these very relevant comments. We agree with the reviewer that Tyr222 is the only Try located near the interface. From the structure information it appears that it is indeed exposed at the outer surface and accessible to the external natural modifiers. In addition, FtsZ(1-366) we used has only two Tyr residues. Another Tyr339 has been shown to have no effects on the pattern changes of FtsZ (doi.org/10.1002/anie.202008691). Furthermore, Tyr222 has also been reported that it can effectively affect GTPase activity and assembly dynamics (doi.org/10.1016/j.freeradbiomed.2017.07.014). We have now discussed this and explained the reason why we choose Tyr222 in the main text (Page 5, Line 21-26).

Page 5, Line 21-26: *We incorporated ClY, Cl₂Y, BrY, Br₂Y, IY and I₂Y at position Tyr222, which is the only tyrosine structurally close to the boundary between the N-terminal domain and C-terminal domain. The position is located on the outer surface of the protein and is exposed to solvent, allowing us to mimic natural halogenation progress that is modified by external natural modifiers. In addition, this position is known to be sensitive to PTMs^{26,46}, making it a good candidate for investigation of halogenations.*

8. I did not understand the term "collective" in the title before reading the text. The authors could reword it.

We have reworded the title to make it easier to understand. *“Halogenation of Tyrosine perturbs large-scale protein self-organization”*

9. Regarding the terms “essential tyrosine” in the title and “at a key position” in the abstract and main text, what makes Tyr222 essential and position 222 a key position? Tyr222 is nothing more than the only Tyr located at the domain interface.

We have rephrased the sentence in the abstract so that it has a clear meaning. (Page 1, line 13-18)

Page 1, line 13-18: *Here, we report a genetically encodable halogenation of tyrosine residues in a reconstituted prokaryotic filamentous cell-division protein (FtsZ) as a platform*

to elucidate the implications of halogenation that can be extrapolated to living systems of much higher complexity. We show how single halogenations can fine-tune protein structures and dynamics of FtsZ with subtle perturbations collectively amplified by the process of FtsZ self-organization.

10. The abstract does not mention FtsZ in it at all. So, the reader can have no idea what the details (structural bending, GTPase activity, etc) in the abstract are all about.

We have now mentioned FtsZ in the abstract to give the reader a better overview of this study (Page 1, line 13-18, see above).

11. “filamenting temperature-sensitive mutant Z” is the name of a mutant. FtsZ is mentioned as a particular protein (the protein product of the *ftsZ* gene) throughout the text. I suggest that lines 70-71 be changed to “...based on the FtsZ protein (the *ftsZ* gene product), the known prokaryotic homologue of the eukaryotic protein tubulin (Fig. 1a).”

We have edited the corresponding text according to the reviewer’s suggestions. (Page 3, line 12-15)

Page 3, line 12-15: *The minimal system studied here²⁴ is a one-component biological system reconstituted in vitro from scratch with the purified FtsZ protein (the *ftsZ* gene product), the known prokaryotic homologue of the eukaryotic protein tubulin (Fig. 1a).*

12. “the bacterial division (Z) ring (l.73) should be changed to “the bacterial division ring (Z ring)”.

We have followed the reviewer’s suggestion and replaced “the bacterial division (Z) ring” with suggested “the bacterial division ring, known as “Z ring”. We think this is now clearer and more professional (Page 3, line 16-18).

Page 3, line 16-18: *As an essential part of the bacterial division ring, known as “Z ring”, FtsZ has shown intriguing self-organization when reconstituted in vitro on biological membranes.*

13. I do not see how the development of halogenated biologics, mentioned in the last part of the discussion, is related to the scope of this study. The authors could discuss this issue at another opportunity.

We thank the reviewer for this suggestion. We have removed the discussion about the development of halogenated biologics from the conclusion section. We now focus mainly on the scope of this study.

Reviewer #2 (Remarks to the Author):

Sun et al. study the effects of tyrosine halogenation of FtsZ on protein self-organisation. For this, experimental and theoretical approaches are combined. The authors state in their conclusion that even a single introduced halogen atom can alter the activity of a protein. The presented results enhance our understanding of oxidative damage-related diseases and enable future applications such as antibody engineering.

While I find the outcome of this study interesting for the general readership of Nature communications, I think the manuscript should be improved so that the non-expert reader can follow the rationale and conclusions drawn from the experiments. Please find some general and specific comments below:

The authors appreciate the reviewer's general impression and supportive comments.

1. The results and discussion are described in one large section. I suggest to provide subheaders to separate the individual experiments and to structure the findings. This will help the reader to follow the results/conclusions.

We thank the reviewer for this suggestion. We have now added subheads to streamline the story and make readers easier to follow.

2. It is difficult for the non-expert reader to understand the principle of the reconstituted minimal cell division system. More explanations would be helpful.

Thanks for pointing out this. We have now reworded the relevant text in the abstract (Page 1, line 13-18) and now described in detail in the introductory paragraph (Page 3, line 12-19). We hope it's now clear to the non-expert readers.

Page 1, line 13-18: Here, we report a genetically encodable halogenation of tyrosine residues in a reconstituted prokaryotic filamentous cell-division protein (FtsZ) as a platform to elucidate the implications of halogenation that can be extrapolated to living systems of much higher complexity. We show how single halogenations can fine-tune protein structures and dynamics of FtsZ with subtle perturbations collectively amplified by the process of FtsZ self-organization.

*Page 3, line 12-19: The minimal system studied here²⁴ is a one-component biological system reconstituted in vitro from scratch with the purified FtsZ protein (the *ftsZ* gene product), the known prokaryotic homologue of the eukaryotic protein tubulin (Fig. 1a). The protein has been shown to be sensitive to halogenating chemicals²⁵ and its activity is sensitive to modifications at individual sites²⁶. As an essential part of the bacterial division ring, known as "Z ring", FtsZ has shown intriguing self-organization when reconstituted in vitro on biological membranes. It polymerizes into dynamic vortices by circular treadmilling dynamics fueled by GTP hydrolysis²⁷.*

3. The latter point is true for most of the described experiments. For instance, it is difficult to understand how results presented in Figure 1c were obtained. Only the figure legend provides information on the experiment that was performed.

We have now detailed our descriptions of experiments in the main text, such as Fig. 1c. To avoid redundancy in the main text, more information about experimental details can be found in the method sections.

Some changes in the main text are listed below:

Page 4, Line 4-9: Halogenation affects several key molecular properties of the parent residue: molecular volume, pKa of the side-chain group, and hydrophobicity (Fig. 1c). We studied the experimental pKa values of the phenolic group in free amino acids, and found that the value depends primarily on the number of halogen atoms introduced to the molecule: Tyr (pKa 9.9) > ClY ≈ BrY ≈ IY (pKa 8.3) > Cl₂Y ≈ Br₂Y ≈ I₂Y (pKa 6.5). Next, we examined

the hydrophobicity of the amino acid residues using lipophilicity measurements using a recently developed method³².

Page 8, line 13-14: We next examined the ring formation behavior of halogenated FtsZ in live E. coli cells (Fig. S5) using fluorescence confocal microscopy.

Page 9, line 5-10: To better understand the architecture and dynamics of the filament network generated by halogenated FtsZ, we reconstituted the protein variants on supported lipid bilayers (SLBs) and quantified their self-organized patterns using a total internal reflection fluorescence microscope (TIRFM) (Fig. 2a, Supplement Movie S1). First, SLB with negatively charged lipid composition was prepared in a home-made microscope chamber. Then FtsZ was introduced on the SLB and self-assembly of FtsZ was initiated by addition of GTP.

Page 10, line 5-7: These proteins display fiber-like and highly meshed filament patterns in the entire membrane area (Fig. 2b). Then we analyzed the curvature of FtsZ filaments with stretching open active contours (SOAX)⁵².

Page 12, line 19-25: We selected two variants: the wild type as the one exhibiting native self-assembly and the Cl₂Y-containing protein as the one unable to form rings. The proteins were mixed in different proportions, with FtsZ(Y222Cl₂Y)-YFP-mts provided at 25%, 50%, and 75%, yielding a total protein concentration of 0.5 μM. The formation of FtsZ ring pattern was examined on SLB under TIRF-monitoring (Fig. 3a, Supplement Movie S2). FtsZ architectures and dynamics, such as ring diameters, velocity and filament curvatures were analyzed.

4. The authors use different abbreviations for the halogenated tyrosine residues (e.g. Cl₁Y, Cl₂Y etc). Abbreviations for the wild-type are mixed throughout the manuscript (Y versus WT versus wild-type etc). I suggest to use the same term for the non-modified protein.

Thanks for your suggestions. To remain consistent, we have now used the same abbreviations in the new version. We used “wild type” throughout the text of the manuscript. To keep labelling short in the Figures, we use the “Y” to indicate wild type. This has been clarified in the figure captions to avoid possible confusion.

5. Schematics of experiments are provided (e.g., Figs 2a and 3a). I suggest to include a short explanation to the text, too.

Thank you very much for the suggestion. We have included brief introductions for the schemes in the manuscript (Page 9, line 5-10, Page 12, Line 19-25).

Page 9, line 5-10: To better understand the architecture and dynamics of the filament network generated by halogenated FtsZ, we reconstituted the protein variants on supported lipid bilayers (SLBs) and quantified their self-organized patterns using a total internal reflection fluorescence microscope (TIRFM) (Fig. 2a, Supplement Movie S1). First, SLB with negatively charged lipid composition was prepared in a home-made microscope chamber. Then FtsZ was introduced on the SLB and self-assembly of FtsZ was initiated by addition of GTP.

Page 12, Line 19-25: *We selected two variants: the wild type as the one exhibiting native self-assembly and the Cl₂Y-containing protein as the one unable to form rings. The proteins were mixed in different proportions, with FtsZ(Y222Cl₂Y)-YFP-mts provided at 25%, 50%, and 75%, yielding a total protein concentration of 0.5 μM. The formation of FtsZ ring pattern was examined on SLB under TIRF-monitoring (Fig. 3a, Supplement Movie S2). FtsZ architectures and dynamics, such as ring diameters, velocity and filament curvatures were analyzed.*

6. How are the mass differences of halogenated tyrosine variants explained? The mass difference is too high. For instance, the mass of chlorine is 35, attachment to tyrosine includes the loss of a H-atom and the observed mass difference should be 34 Da.

We appreciate the concern raised by the referee that points out an important issue. In this study, the molecular weights of halogenated protein were determined with Electrospray Ionisation Mass Spectrometry (ESI-MS). The accuracy of ESI-MS for protein molecules has been documented to be ± 0.01% compared with the theoretical amino acids sequence (*Clin Biochem Rev.* 2003;24(1):3-12.; doi.org/10.1016/0076-6879(90)93432-K). Therefore, the observed mass in our study has some discrepancies that can be attributed to the accuracy of ESI-MS. Therefore, we cannot determine the mass difference of halogenated protein as accurately as the theoretical values. However, it should be noticed that all our measurements are within the accuracy range of measured results are reliable and fully reproducible. We have explained more in the heading of Supplement Table S3 to avoid possible confusions.

*TableS3 caption SI-Page 24, Line 3-4: *The accuracy of ESI-MS for protein molecules has been documented to be ± 0.01% when compared with the theoretical amino acids sequence^{40, 41}.*

7. Which mass spectrometer was used? The methods sections states that a Q-ToF instrument from Agilent was used, but the proteins were introduced into a mass spectrometer from Thermo Scientific. The analysis was performed with an Orbitrap but a Q-ToF was used. It is unclear how the experiments were performed. Mass spectra shown in Fig. 1d do not show the expected high resolution of an Orbitrap mass analyser. In addition, experimental details are missing (e.g. specification of the column used (particle size etc), parameters of the MS analysis (in particular ion mode)). The MS section needs to be corrected.

We thank the reviewer for these important technical issues. These remind us that our method sections were not clear enough. In the new version, we have detailed the mass spectrometry method section with all the necessary information, including the type of mass spectrometer, method of analysis, and other MS parameters (Page 21, line 15-24)

As for the mass spectra in Fig. 1d, the peaks are broad and not sharp, mainly because of the preparation of the figure. In the previous version, we made the x-axis area too small, compressed the height of the figure, and zoomed in too much to show the difference in the mass peak. In the new version, we optimized the visualization of Fig. 1d and remeasured the mass spectra, e.g., FtsZ(Y222Cl₂Y)-YFP-mts and FtsZ(Y222Cl₂Y)-YFP-mts, to improve their quality. We also provide an additional supplement figure(S4) in the SI to demonstrate the spectrums

Page 21, line 15-24: *Mass-spectrometry Intact mass measurements of purified proteins were performed by electrospray LC-MS on an Agilent 6530 QTOF instrument coupled with an*

Agilent 1260 HPLC system after external calibration. 80-100 μ L of a protein solution with a concentration around 0.1mg/mL was prepared. Samples were infused at a flow rate of 0.3 mL min⁻¹ onto a gradient from 5 % acetonitrile 0.1 % formic acid in water to 80 % acetonitrile 0.1 % formic acid in water through a C5 column, 2.1x100 mm, 3 micron (Supelco analytical, Sigma-Aldrich) over 20 minutes. The protein was ionized via electrospray ionization (ESI). Spectra deconvolution was performed with Agilent MassHunter Qualitative Analysis software (v. B.06.00, Bioconfirm Intact mass module) employing the maximum entropy deconvolution algorithm.

Reviewer #3 (Remarks to the Author):

In their manuscript Sun et al. describe the effect of tyrosine halogenation on FtsZ polymerization. This is a very interesting system to study, because FtsZ is essential in prokaryotic cell division and acts as interaction hub to recruit other cell division proteins. The authors show that mutation (halogenation) of a single residues in FtsZ causes significant phenotypic changes. The use of genetically encoded halogenated amino acids is a nice demonstration of this powerful technique to produce homogeneous samples and cleanly analyze their function. The self-organization assays are well executed and are informative. Basic molecular dynamics simulations are presented, but the complexity of the system make the results hard to interpret. My main criticism with the manuscript is that it mainly presents observations but falls, in its current form, short in clearly rationalizing and explaining results. The authors acknowledge this themselves by stating “Because the physiological complexity it is not yet possible to derive a particular underlying mechanism from these cellular results. However, the results are sufficient to ascertain dramatic effects resulting from single site halogenation”. Overall I believe the manuscript could be improved by the authors addressing the following points and revising the manuscript accordingly:

We appreciate referee comment and agree that a better presentation can be provided. In the revised manuscript, we have done our best to explain our results, especially in the discussion of GTPase activity (Page 6, Line 15-21), *in vivo* results (Page 8, line 22-Page 9 line 4), and structural simulation (Page15, line 19-26). In discussing the *in vivo* results, we found that the halogenation can disrupt Z ring formation in *E. coli* and inhibit treadmilling dynamics. However, due to the physiological complexity and high background of endogenous FtsZ, it is not yet possible to deduce a specific underlying molecular mechanism and the specific effects of different types of halogenations from these cellular results (Page 8, line 22-Page

9 line 4). Then we performed the *in vitro* experiments because they gave us cleaner results. We have now clearly stated this in the main text.

1. on page 5 it is stated “Thus we infer that tyrosine halogenation may dramatically affect sidechain acidity, volume and hydrophobicity, while these perturbations subsequently lead to suppressed GTPase activities.” The authors should correlate physicochemical properties of the noncanonical amino acids (Figure 1c) with their measured rates of GTP hydrolysis and rates of polymerization (Figure 1f). Neither of these properties seem to correlate with rate of hydrolysis or polymerization.

We appreciate the concern raised by the referee that points out an important issue. We have now interpreted our results of GTP hydrolysis and polymerization rates in the context of the physicochemical properties of unnatural amino acids in Fig. 1c, when we talked in general terms about the suppressed GTP hydrolysis by halogenations (Page 6, line 15-21). However, when comparing the functions of different halogenated Tyr analogues (HYs), it is still not possible to draw a direct correlation. There is clearly no correlation with side-chain pKa, volume, or hydrophobicity when analysing only one factor. We thought the overall outputs might be affected by interaction of two or more factors and their interaction with the environment of the proteins. Therefore, we cannot draw a definite conclusion at the stage of GTPase measurement. However, when discussing protein assembly, we gave some interpretations showing how the physicochemical properties of HYs are related to protein pattern formation (Page 9, line 17-23, Page 10 line 14-24).

Page 6, line 15-21: *We infer that the molecular perturbations could result from the changes of key molecular properties demonstrated in Fig. 1c, such as side-chain acidity, molecular volume, and hydrophobicity. As mentioned before, the pKa values of HYs at pH7.5 were all lower than that of tyrosine and the lipophilicity was higher than that of tyrosine, which could increase the acidity of tyrosine side-chain and hydrophobicity, leading to suppressed GTPase activities. In addition, halogenation can enlarge the volumes of the amino acids, and thus might disrupt the protein structures.*

2. The selection of tRNA synthetase enzymes for the noncanonical amino acids seems unusual, because only one round of positive selection was used. Previous studies using variant libraries of the *M. jannaschii* tRNA synthetase report that the gene pool after the first positive selection round mainly contains RS enzymes that recognize canonical amino acids. A more detailed description should be include, because there is clearly important information missing. The authors should clearly state how many colonies were screened and found selective for noncanonical amino acids. Further, it should be clearly stated which plasmids are used for different experiments. I.e. “genes of interest were cloned into pULTRA for further experiments and/or directly applied to a 96-well fluorescence assay”. Is Figure 2b produced with pULTRA or pBU16 plasmids?

We appreciate this relevant question. Indeed, in this research we used a simplified one-round selection approach without negative selection, according to the reported method (doi.org/10.1002/ange.201400001, doi.org/10.1038/nchembio.2406). For selection, we used chloramphenicol acetyltransferase with two amber codons and a small ubiquitin-like

modifier tagged with superfolder green fluorescent protein (sfGFP) with one in-frame amber stop codon for readout. Compared to the traditional selection approach with positive and negative selections, one selection round with two readout is already sufficient and eliminates the need for negative selection. This method has become established as a standard approach, is frequently used in practice, and has proven as an efficient selection method. To avoid possible confusion, we have now described the method in detail in both the main text and the methods section. Additional references are cited. The number of colonies and the efficiency of selection are now given in the methods section (Page 19, line 3-11). To address the reviewer's concern about the plasmids used, we have included the relevant information in the methods section (Page 19, line 3-11; Page 19, line 18-21; Page 20, line 26-Page 21, line 4) and supplied a supplementary table (S8), e.g., the pULTRA plasmid was used to generate halogenated FtsZ in Fig.2.

Page 19, line 3-11: *37 out of 100 clones showed Br₂Y-dependent survival and green fluorescence. Meanwhile, 27 out of 66 colonies can grow in the presence of Cl₂Y and generate green fluorescence (Fig. S2a). Nine selected clones containing unique synthetase genes of interest were directly applied to a 96-well fluorescence assay to test the incorporation efficiency or poly-specificity of screened MjTyrRSs. The selected MjTyrRSs specific for HYs were then cloned into pULTRA⁴ for halogenated FtsZ production.*

Page 19, line 18-21: *Then clones containing the 9 unique MjTyrRSs (pBU18'1GK plasmid) and positive selection plasmid (pPAB26_cat (Q98TAG, D181TAG) MjtRNATyrCUA-his-SUMO-sfGFPR2TAG-strep), which selectively grew on the plates in the presence of Br₂Y or Cl₂Y were directly cultured in LB medium with Amp and Kan at 37°C, overnight.*

Page 20, line 26-Page 21, line 4: *The selected synthetase B48RS and C64RS were cloned into pULTRA plasmid. This highly efficient suppressor plasmid, pULTRA, harbors a single copy each of the tRNA and aaRS expression cassettes that exhibits higher suppression activity than its predecessors... The gene of FtsZ(Y222TAG)-YFP-mts containing an amber codon at Y222 position was cloned on pET-28a expression vector and co-transformed with pULTRA_B48RS_tRNATyrCUA (for Cl₁Y, Cl₂Y, Br₁Y, Br₂Y, and IY incorporation) or pULTRA_C64RS_tRNATyrCUA (for I₂Y incorporation) into RF1 free BL21(DE3) (B95.ΔA)⁵.*

Table S8. List of plasmids used in this study

Plasmid	Replication Origin	Resistance	Description
pBU18'1GK_Library MjTyrRS	pUC	AmpR	MjTyrRS Library (Fig.S1, Fig.S2)
pPAB26_cat (Q98TAG, D181TAG) Mj tRNA ^{Tyr} CUA-his-SUMO-sfGFP ^{R2} TAG-strep	p15A	KanR	One-round Selection plasmid (Fig.S1, Fig.S2); SUMO-sfGFP production (Fig. S3)
pBU18'1GK_B4RS(B4RS)	pUC	AmpR	
pBU18'1GK_B12RS(B12RS)	pUC	AmpR	
pBU18'1GK_B37RS(B372RS)	pUC	AmpR	tRNA Synthetase (Fig.S1, Fig.S2);
pBU18'1GK_B39RS(B39RS)	pUC	AmpR	B48RS and C64RS
pBU18'1GK_C5RS(C12RS)	pUC	AmpR	were used for SUMO-sfGFP production (Fig. S3)
pBU18'1GK_C45RS(C45RS)	pUC	AmpR	
pBU18'1GK_C61RS(C61RS)	pUC	AmpR	
pBU18'1GK_B48RS(B48RS)	pUC	AmpR	
pBU18'1GK_C64RS(C64RS)	pUC	AmpR	
pULTRA_B48RS_tRNA ^{Tyr} CUA	CloDF13	SpecR	Halogenated FtsZ Protein expression (Fig.1.d.e.f, Fig.2, Fig.3 Fig.S4)
pULTRA_C64RS_tRNA ^{Tyr} CUA	CloDF13	SpecR	
pET28a_FtsZ (Y222TAG)-YFP-mts	ColE1	KanR	
pET11b_wt_FtsZ-YFP-mts*	ColE1	AmpR	Wild type FtsZ protein expression (Fig.1.d.e.f, Fig.2, Fig.3 Fig.S4)

AmpR: Ampicillin resistance; KanR: Kanamycin resistance; SpecR: Spectinomycin resistance. *Ramirez-Diaz DA, Garcia-Soriano DA, Raso A, Mucksch J, Feingold M, Rivas G, Schwille P: **Treadmilling analysis reveals new insights into dynamic FtsZ ring architecture.** *PLoS biology* 2018, **16**:e2004845.

- The quality of the mass spectrometry data is on the lower end of what can be expected from the specified setup. While the peak maxima correspond well with the expected masses, the mass signals are broad and not resolved. This may be a setting in the deconvolution program. Figure 1d clearly shows satellite peaks for Cl₂Y and I₂Y indicating dehalogenation and there are higher mass peaks consistent in all samples. Better mass spectra quality likely reveal their identity, which may also be important in the discussion of effects on function. The mass spectra in Figure S3 would be more useful if only the mass range of 40000 to 41000 Da was shown in high resolution.

We are grateful for this very interesting and relevant comment. The peaks are wide and not sharp, which is mainly due to the preparation of the figure. In the previous version, we chose the x-axis area too small, compressed the height of the figure, and zoomed in too much to show the differences in the peak mass. In the new version, we optimized visualization of the Fig. 1d and remeasured the mass spectra, e.g., FtsZ(Y222CIY)-YFP-mts and FtsZ(Y222Cl₂Y)-YFP-mts, to improve their quality. We have also provided an additional supplementary fig. S4 in the SI to clarify the satellite peaks in the spectra. In addition, the mass spectrum of FtsZ(Y222Cl₂Y)-YFP-mts becomes cleaner when measured with fresh samples and the smaller satellite peak is no longer shown in the new data. We also measured the FtsZ(Y222I₂Y)-YFP-mts but could not get better and cleaner results. The satellite peak is still present. The smaller peak actually indicates the dehalogenation, as the reviewer suggested. This is because the formic acid can induce dehalogenation of the iodinated aromatic compounds during ESI-MS measurement (doi: 10.1002/rcm.6711). The larger satellite peaks are sodium species adducts. We have now clarified these concerns in the caption of Supplementary Fig. S4. In Fig. S3, we fix the mass range from 40000 to 41000Da and believe that it is now better.

Fig. 1d

Figure S4. The deconvoluted and non-deconvoluted ESI-MS spectra of FtsZ-YFP-mts incorporated with HYs. The observed and expected molecular masses are as follows: a.) wild type FtsZ-YFP-mts (Y): main peak (observed: 68060.71 Da, expected: 68061.29 Da). b.) FtsZ(Y222ClY)-YFP-mts: main peak (observed: 68094.02 Da, expected: 68095.73 Da); c.) FtsZ(Y222Cl₂Y)-YFP-mts: main peak: observed: 68130.04 Da, expected: 68130.29 Da. d.) FtsZ(Y222BrY)-YFP-mts: main peak (observed: 68139.11 Da, expected: 68140.18 Da). e.) FtsZ(Y222 Br₂Y)-YFP-mts: main peak (observed: 68219.41 Da, expected: 68219.08 Da). f.) FtsZ(Y222IY)-YFP-mts: main peak (observed: 68186.04 Da, expected: 68187.19 Da). g.) FtsZ(Y222I₂Y)-YFP-mts: main peak (observed: 68312.59 Da, expected: 68313.08 Da). All the satellite peaks are listed below: Peak1 (67685.53 Da), Peak2 (67719.94 Da), Peak5 (67765.03 Da), Peak6 (67844.63 Da), Peak7 (67808.00 Da) and Peak8 (67937.56 Da) represent the proteins that are around 375 Da smaller compared to the main peak, which probably resulted from the loss the of C-terminal α -helix of FtsZ-YFP-mts during ESI-MS measurement. The mts from MinD is a positively charged domain that is unstable under the acidic conditions applied in the ESI-MS measurement protocol due to a low ionic strength²³. It has been reported that the N-terminal or C-terminal α -helices can be prone to a loss under low ionic strength²⁴. Peak3 (observed: 68027.10 Da, expected: 68,027.52 Da) indicates FtsZ(Y222ClY)-YFP-mts without N-terminal Met but with 3 Na⁺ adduct. Peak4 (observed: 67652.90 Da, expected: 67,652.52 Da) represents FtsZ(Y222ClY)-YFP-mts without N-terminal Met and C-terminal mts (during ESI-MS measurement) but with 3 Na⁺ adduct. Peak 9 (observed: 68186.01 Da, expected: 68186.69 Da) is deiodination of FtsZ(Y222 I₂Y)-YFP-mts caused by formic acid during MS measurement²⁵.

4. The addition of homology modelling and molecular dynamics simulations appears to be an add-on. Details on how the MD simulations were performed are largely missing and cannot be reproduced from the current description. Molecular dynamics simulations are difficult to interpret at the best of times, and I would suggest not to rely on single trajectories for interpretation.

Thank you very much for your comments. Homology modelling was performed in this work as a solution to the absence of structural data of the FtsZ protein from *E. coli*. The homology models were thermally equilibrated at 300 K for 10 ns following the protocols described under “FtsZ structure simulation” in SI. The description of the molecular modelling and simulations has been reorganized in main text (Page 14, line 10-26) and SI to help the readership understand our computational approach. The protocol used for MD simulation has been extended in SI. (SI-Page 9, line 20-Page 10, line 8).

Page 14, line 10-26: *Structure modelling elucidates the effects of halogenation*

*To better understand how the subtle changes in halogenated protein structure suppress protein activity and how the effects are further amplified by the self-organization process, we constructed structural models for the wild type FtsZ monomer and dimer, as well as for the FtsZ(Y222Cl₂Y) mutant including a GTP molecule at the interface and in the upper monomer binding site. Starting coordinates were generated from a homology model of the wild type FtsZ with the I-TASSER suite³³ using the *M. jannaschii* FtsZ dimer structure (PDB code: 1W5A)⁴⁰(Fig. 4a, Fig. S6) as template. Compared to the published structures of the two monomer conformations⁴¹, the structural models were identified to be in the closed state i. e., a conformation preferred in solution. Thus, the FtsZ dimer models represent the*

nucleation of a filament. Structures containing the mutated residues FtsZ(Y222Cl₂Y) were modelled by modifying the tyrosine residue in the homology model. Briefly, chlorine positions were taken by aligning the modelled residue with a crystal structure of a halogenated tyrosine (PDB: 4NX2)⁵⁴ (Fig. 4b) and partial charges for the backbone were calculated (Table S5). Detail description of the molecular modeling protocol is found in supplementary information. These models of wild type and halogenated FtsZ monomers and dimers, were subjected to 500 ns molecular dynamics (MD) simulations (see SI) to gain insight into protein dynamics and GTP binding.

SI-Page 9, line 20-Page 10, line 8: *Molecular dynamics simulation*

In this study, all molecular dynamics (MD) modelling was performed using the Amber software suite⁷, using the recommended ff14SB, TIP3P and gaff2 force fields for protein, water, and generalised molecules respectively¹⁵ and in-house developed GTP parameters¹⁶. In each simulation, the protein was solvated in a TIP3P¹⁷ periodic box, measuring 12 Å from the edges of the protein. The system was neutralized with K⁺ ions, which were distributed based on a Coulombic potential grid. No further ions were added, apart from the Mg²⁺ ions associated with the active site. The energy of the system was minimized for 12 000 steps, the first 10 000 steps using the steepest descent method. The system was then heated from 0 K to 300 K over the course of 500 ps using an NVT ensemble, during which backbone atoms were constrained. Langevin dynamics were used to control the temperature with a collision frequency of 2.0 ps⁻¹¹⁸. The system was equilibrated for 200 ps without constraints on backbone atoms preceding production. A 2 fs integration time-step was used for the simulation and the SHAKE algorithm¹⁹ was employed to constrain hydrogen bonds during the simulation. Non-bonded terms were cut off at 12 Å; particle mesh Ewald (PME) was used to treat long-range electrostatics²⁰. Simulations were performed on a CUDA-accelerated platform with 12 cores at 3.0 GHz and 128 GB RAM.

5. Throughout the manuscript the authors should check the precision with which experimental results are reported, and only report significant figures. I.e. In Figure 4c, are the binding energies really accurate to 0.01 kJ/mol? Provide error estimates. Observed protein masses are reported with 2 significant digits.

We thank to the referee for bringing this issue to our attention. Our binding energy results are accurately presented without decimal points and we have now included this in the manuscript. These changes do not affect the observed trends and claims related to GTP-hydrolysis. Our electrostatics calculations were performed with the highest possible precision (grid potential resolution of 0.5 Å) on available computers. For such a large system, the RAM memory requirements increase exponentially at higher grid resolutions in the APBS suite [N. A. Baker, D. Sept, S. Joseph, M. J. Holst, J. A. McCammon, Proc. Natl. Acad. Sci. U. S. A. 2001, 98, 10037–41.]. We also checked and corrected the accuracy of protein mass, ring size, and ring velocity throughout the manuscript.

Reviewer #4 (Remarks to the Author):

Sun et al. presented the experimental platform along with the evidence that halogenation of a particular tyrosine residue in FtsZ at a molecular-level perturbs the self-organization of FtsZ polymers at a cellular-level. The study combined *in vivo* live-cell imaging, *in vitro*

reconstitution, and molecular dynamics simulations to substantiate their finding and conclusions. Since halogenation is an important form of non-enzymatic post-translational modifications with the latter being an emerging factor in several oxidative stress-related diseases, biophysical and biochemical characterizations of the halogenation effects in this study could be of potential importance and interest in the general field of biophysics and synthetic biology. That being said, I'd be delighted to see this work in publication. However, I'd suggest the authors to consider the following suggestions to improve their work/manuscript before publication.

The authors thank the reviewer's supportive feedback and helpful suggestions.

1. The authors elucidated the halogenation effects on the formation and GTPase-mediated treadmilling dynamics of FtsZ ring with the *in vitro* reconstitution. The biophysical and biochemical characterizations seem to be solid. But if I understand correctly, I failed to see how this particular effect on FtsZ has anything to do with the general principle of protein halogenation. Is the halogenation effect just another post-translational modification or unique in some aspect(s)?

We thank the reviewer for these useful comments. In this research, we take the minimal system of FtsZ as a platform to investigate the general effects of halogenations on protein activities, protein dynamics and self-assembly. We found that the changes in hydrophobicity, volume, and acidity induced by halogenations affect enzymatic activities and alter protein structures in a subtle way, which may explain the general effects of halogenations on other proteins. In addition, we use the filamenting protein (FtsZ) to interpret how a single halogen atom suppresses protein activities and how the effects are amplified during protein assembly. The present study provides general insights into the amplification effects of halogenation or other PTMs that could be applied to the self-assembly of other proteins, especially cytoskeletal proteins.

Regarding the properties of halogenations, they are not just PTMs allowing us probing their roles in protein structure and function but may also be unique tools for fine-tuning protein structures and functions. With their broad *pKa* spectrum, hydrophobicity, and slightly altered volume size, halogenated tyrosine analogues could be a novel regulatory toolbox that meets the diverse needs of synthetic biology. We now addressed all the general insights in the conclusions (Page 16, Line 12-26).

Page 16, Line 12-26: *Through the in vitro protein assay and theoretical structure simulation, we discovered that even one or two newly introduced halogen atoms could readily alter the enzymatic activity of the protein, amplify or propagate their effects through protein-protein interactions, and subsequently produce global effects on protein pattern formation and dynamics. The in vitro methodology used here is not limited to a quantitative and conceptual understanding of halogenation, but could also be applied to other types of post-translational modification occurring in natural proteins. Furthermore, as a kind of fine-tuning tools⁵⁶ our halogenated tyrosine enables the study of complex protein systems in vitro and in vivo with residue precision. The different variants of halogen atoms and modified positions will allow production of a large repertoire of new modification functions in the future. Specifically, HYs are typical products of the oxidation of the tyrosine residue in proteins, ultimately contributing to aging processes³. The site-specific modification of protein structures and activities in the presence of HYs should therefore enable a much better understanding of the*

oxidative damage-related diseases such as aging and cancer. Accumulation of this information will further help improve diagnostic methods and therapeutic interventions for patients in the future⁵⁷⁻⁵⁹.

2. In light of FtsZ, the central organizer of bacterial divisome, the authors demonstrated that different kinds of halogenations at Y222 can change the geometry of FtsZ ring and slow the treadmill speed. I have several questions as follows.

2.1) The treadmill speed of the wt FtsZ-YFP-MTS is ~ 20 nm/s, which is much slower than those *in vivo* (~ 30-40 nm/s). What causes this slower speed? Does it relate to the fact that FtsZ is linked to membrane by MTS (membrane target sequence), rather than ZipA and FtsA as that *in vivo*? If so, then how will the halogenation effect play out *in vivo* with the presence of ZipA and FtsA? Is it possible that the halogenation effects are masked by ZipA and FtsA so that treadmill speed may not be perturbed? The authors need to demonstrate the relevance of halogenation effects on FtsZ treadmill speed *in vivo*.

We thank the reviewer for bringing this interesting point to our attention. We agree that the slower speed might be caused by different anchors. Loose group used FtsA as anchor instead of MTS and showed the velocity of FtsZ is around 33 nm/s, which is close to the *in vivo* results mentioned. (doi.org/10.1038/s41564-019-0657-5). However, Ramirez-Diaz et. al use MTS as anchor and show the velocity is around 25 nm/s (doi.org/10.1371/journal.pbio.2004845), which is closed with our results (22±5 nm/s). Although the velocity is slightly dependent on the difference of anchors, it is known that the velocity reflects the intrinsic properties of FtsZ (doi.org/10.1371/journal.pbio.2004845), such as GTPase activity, when the same anchor is used. Once the type of anchor is determined, halogenation will mainly influence intrinsic properties of FtsZ. Since different anchors will not affect the intrinsic properties of FtsZ, they will also not mask the effects of halogenation.

We also thought it would be interesting to study the dynamics of halogenated FtsZ in *E. coli*. However, there are some challenges for investigation of single atom modification *in vivo* now. 1. The background of the *in vivo* study is extremely high due to the complexity of the system and endogenous FtsZ of *E. coli*. Therefore, the subtle changes in FtsZ structures (overexpressed) after halogenation may not be apparent, since there are endogenous FtsZ and many FtsZ-associated proteins involved. Thus, the results will be not clean enough to distinguish the difference between different halogenations (see below the new FRAP results). 2. It's still a challenge to specifically modify the endogenous FtsZ. Thus, clean measurement of FtsZ treadmill in *E. coli* requires completely new experimental setups and hardware, which cannot be achieved in a short time. We think this will be a separate narrative that will be explored in future research.

To address the reviewer's concern as much as we can, we performed photobleaching recovery (FRAP) experiments that can reflect the treadmill dynamics of halogenated FtsZ *in vivo* and help to understand the importance of halogenation effects on FtsZ treadmill speed *in vivo*. (doi.org/10.1073/pnas.052595099). The results are shown in Figure.S5h-I and discussed in the main text (Page 8, Line 24-Page 9, line 4; Page 9, Line 26-28; Page 10, Line 12-14).

Page 8, Line 24-Page 9, line 4: *To investigate the dynamics of FtsZ assembly in vivo, we performed fluorescence recovery after photobleaching (FRAP) analysis on these FtsZ*

architectures. In FRAP analysis, a portion of a fluorescent FtsZ pattern is photo-bleached with a focused laser beam and the recovery of the fluorescence signal caused by replacement of subunits outside the photobleached region is measured. We found that the fluorescence recovery of wild type FtsZ-YFP-*mts* was faster than that of the halogenated FtsZ-YFP-*mts* (Fig. 5h-i), which is consistent with the trend of GTPase activity shown in Fig. 1e. All of the above results are sufficient to ascertain effects on both structures and dynamics of FtsZ resulting from single site halogenation. However, because of the physiological complexity and high background of endogenous FtsZ, it is not yet possible to deduce a specific underlying molecular mechanism and the specific effects of the different types of halogenations from these cellular results.

Page 9, Line 26-28: Compared with the more noisy and complex *in vivo* approaches (Fig. S5), we expected treadmilling dynamics to provide a clearer illustration of the protein damage, since this assay is performed purely *in vitro*.

Page 10, Line 12-14: Conversely, with Cl₂Y and IY containing proteins the half-time was nearly doubled, indicating slower dynamics (Fig. 2f, g and Table S4). The lowered dynamics comes in agreement with the *in vivo* results (Fig. S5 h-i).

Figure S5 h-i) FRAP analysis of dynamics of FtsZ in *E. coli*. h.) Representative snapshots of *E. coli* cell overexpressed with wild type FtsZ-YFP-*mts* and FtsZ(Y222IY)-YFP-*mts* after photo-bleaching. Scale bar: 1 μm. i.) Fluorescence recovery curves of wild type FtsZ-YFP-*mts* and halogenated FtsZ(Y222X)-YFP-*mts* (X = ClY, BrY, Br₂Y, I₂Y, Cl₂Y, and IY) after photo-bleaching.

2.2) It is difficult to discern the FtsZ ring structures in Cl₂Y and IY (figs 2b and 3b); instead, they look to me more like a cluster without the close loop. Why did author term these structures as rings? How did the authors quantify the ring diameter? Will it be more meaningful to quantify the length of FtsZ filament length in these clusters?

We appreciate these relevant questions. Indeed, there are no distinctive ring formations for FtsZ Containing Cl₂Y and IY, since the two can only form the fibre like meshwork in Fig. 2b. We didn't term them as rings in the manuscript. "Page 10, line 3-6: In two cases, the

presence of HYs resulted in full suppression of the ring pattern formation *in vitro*. We did not observe any distinctive ring formation produced by protein variants containing Cl₂Y and IY.” In Fig. 3b, ring patterns can be seen because we mixed the wild type FtsZ with halogenated FtsZ. But the ring can only form at a certation ratio of wild type. The lower the ratio of wild type, the more similar the pattern becomes to the patterns of pure Cl₂Y with FtsZ.

Quantification of filament length could be an option, but direct comparison of the mature patterns of wild type and (Cl₂Y and IY) is not fair because they form different patterns, one is ring, another is meshwork. Moreover, in the mature pattern, it is difficult to judge the length of each filament. Before pattern maturation and in the early stage of filament formation, filament length varied during assembly and therefore provides little insight into protein pattern formation. Therefore, instead of lengths, we calculated the overall curvature of the filaments of Cl₂Y and IY, which we found more useful for understanding the observed phenomenon.

2.3) If the FtsZ ring is ~ 0.8-1 micron in diameter, shouldn't the curvature of the ring be ~ $1/0.4 \sim 1/0.5$ (micron⁻¹), which is ~ 2 – 2.5 micron⁻¹? However, the figs. 2f and 3d show that the curvature, if I understand it correctly, is on the order of 0.01 micron⁻¹.

We appreciate the reviewer's comments, which makes us realize that the information is not sufficient clear in the main text. As mentioned in question 2.2, we did not observe ring formation for FtsZ containing Cl₂Y and IY. Therefore, the filament (not ring) curvature of FtsZ bearing Cl₂Y and IY was quantified in Fig. 2f. For the curvature results in Fig. 3d, the curvatures for the mixture of ring and filaments were averaged, representing the overall curvatures of the bulk reaction. Thus, the rough estimation of the bulk reaction is not comparable to the theoretical calculation of a single ring. To make this clear, we have reworded the corresponding text in the caption of Fig. 3 (Page 13, Line 9-10).

Page 13, Line 9-10: *Curvatures in d. were calculated for the mixture of ring and filaments, representing the overall curvatures of the bulk reaction.*

3. The authors leveraged molecular dynamics simulation (by AMBER) to study how halogenations affect the energy and dynamics of FtsZ dimerization. It is intuitive that if the FtsZ dimer has a sharper kink, then a smaller FtsZ ring tends to form (aka with a higher curvature). I have two questions.

3.1) The simulation run is ~ 100 nanoseconds, and yet, the FtsZ ring formation and treadmilling is on the timescale of seconds to minutes. There is a gap between the MD simulation result and the experiment observation on cellular scales; this always leaves the interpretation of MD simulation result in some limbo. Can the authors elaborate what are expected to hold from their MD simulation result on the cellular scales? And what are not?

We appreciate the comment on the simulation length. The processes described in this work, self-assembly of FtsZ and filament formation, occur on time scale significantly longer than the 100-ns simulation time and involve much more than two FtsZ monomers. In our computational work, we tried to generate a model to specifically describe only the nucleation event involving two monomers in a filament. Therefore, we studied conformational dynamics of the dimers and monomers as well as GTP-binding to interpret the possible behaviour of these constructs in wild type and after halogenation. We agree that

our results cannot explain the entire treadmilling process at the cellular level, but they provide insight into the conformational differences after tyrosine halogenation, which serves as a proof of concept for experiments that the self-assembly likely takes a different route in wild type. Additionally, we believe that a fairly reliable interpretation can be made when observing mentioned conformational differences on time scales in this work.

3.2) Why did the authors choose the dielectric constant to be 4? To put this in context, the dielectric constant is 80 for water and 2-3 for lipid bilayer/membrane. Given FtsZ polymer is periphery to membrane and hence in cytosol, the interactions within and between the individual FtsZ dimer are expected to be in water. Setting the dielectric constant to be 4, did the authors imply that the FtsZ dimer in membrane? Or is this dielectric constant referred as the internal dielectric constant within the protein? If so, then what is the physical underpinning of choosing this internal dielectric constant to be 4? More importantly, how does the simulation results in figs. 4c, f, and i change with the value of this dielectric constant?

We appreciate the concern raised by the referee. We understand that the dielectric constant of proteins can vary. It can be as low as 2 for very dry proteins and as high as 20 for highly solvent-exposed protein parts. This has already been discussed in references [1, 2, 3]. Here we do not assume that FtsZ is in the membrane, but rather in solution and the value of 4 refers to the internal protein dielectric constant. In all our computations the solvent was represented as a volume with a dielectric constant of 80. The choice of 4 as protein dielectric constant allows for proper treatment of small backbone fluctuations and polarization, the effects which would lower the dielectric constant significantly if treated explicitly [1, 2, 4-7]. Additionally, using this value has been proven to give good agreement with experiments when comparing electrostatic properties [4, 8, 9]. In a previous work, we performed computations with adjacent values of 3 and 6 [10] which did not change the observed trends and conclusions regarding pKa values computed with electrostatic energy computations (although the results were numerically different, as expected). We have now explained in more detail in the caption of Figure S8 in the SI (SI-Page 20, Line 10-14).

[1] Warshel, A. and Papazyan, A. *Electrostatic effects in macromolecules: fundamental concepts and practical modeling. Current Opinion in Structural Biology* 8:211–217, 1998.

[2] Schutz, C. N. and Warshel, A. *What are the dielectric "constants" of proteins and how to validate electrostatic models? Proteins: Structure, Function, and Bioinformatics* 44:400–417, 2001.

[3] Li, L.; Li, C.; Zhang, Z.; Alexov, E. *J. Chem. Theory Comput.* 2013, 9, 2126.

[4] Rabenstein, B., Ullmann, G. M., and Knapp., E.-W. *Energetics of electron-transfer and protonation reactions of the quinones in the photosynthetic reaction center of rhodospseudomonas viridis. Biochemistry* 37:2488–2495, 1998.

[5] Rosen, D. *Dielectric properties of protein powders with adsorbed water. Trans. Faraday Soc.* 59:2178–2191, 1963.

[6] Gilson, M. K. and Honig, B. *Calculation of the total electrostatic energy of a macromolecular system: Solvation energies, binding energies, and conformational analysis. Proteins: Structure, Function, and Bioinformatics* 4, 1988.

[7] Gilson, M. K. and Honig, B. H. *The dielectric constant of a folded protein. Biopolymers* 25:2097–2119, 1986.

[8] Baker, N., Sept, D., Joseph, S., Holst, M., and McCammon, J. *Electrostatics of nanosystems: application to microtubules and the ribosome. Proc. Natl. Acad. Sci. U.S.A.* 98:10037–10041, 2001.

[9] Ishikita, H., Morra, G., and Knapp, E.-W. *Redox potential of quinones in photosynthetic reaction centers from rhodobacter sphaeroides: Dependence on protonation of Glu-L212 and Asp-L213. Biochemistry* 42:3882–3892, 2003.

[10] Batebi, Hossein, Jovan Dragelj, and Petra Imhof. "Role of AP-endonuclease (Ape1) active site residues in stabilization of the reactant enzyme-DNA complex." *Proteins: Structure, Function, and Bioinformatics* 86.4 (2018): 439-453.

SI-Page 20, Line 10-14: *Here we do not assume that FtsZ is in the membrane, but rather in solution and the value of 4 refers to the protein dielectric constant. The protein dielectric value of 4 was chosen in order to allow proper treatment of polarization and small backbone fluctuations, the effects which would lower the dielectric constant significantly if treated explicitly²⁵⁻³⁰.*

REVIEWERS' COMMENTS

Reviewer #1 (Remarks to the Author):

All my concerns have been addressed successfully. I believe that the manuscript is worth publishing in its current form.

Reviewer #2 (Remarks to the Author):

The authors addressed all my comments sufficiently. I recommend publiaction of their revised manuscript.

Reviewer #3 (Remarks to the Author):

The authors have made appropriate changes to the manuscript according to my comments, and I'm satisfied with their changes.

However, the authors also added to the manuscript a section to justify the low mass accuracy (and resolution) in their mass spectrometry results. In their support, the authors cite a 30+ year old reference (and a 20 year old paper that cites this reference). I would advise not to use these outdated references in Table S3, because a modern mass spectrometer, like the one used in this study, can achieve easily mass accuracies of less than 5 ppm.

Reviewer #4 (Remarks to the Author):

The authors have adequately addressed my questions/concerns. I recommend this manuscript for publications.

We appreciate all the efforts from the reviewers and editors.

Reviewer #3 (Remarks to the Author):

The authors have made appropriate changes to the manuscript according to my comments, and I'm satisfied with their changes.

However, the authors also added to the manuscript a section to justify the low mass accuracy (and resolution) in their mass spectrometry results. In their support, the authors cite a 30+ year old reference (and a 20 year old paper that cites this reference). I would advise not to use these outdated references in Table S3, because a modern mass spectrometer, like the one used in this study, can achieve easily mass accuracies of less than 5 ppm.

We thank the reviewer for pointing out the outdated references. These are indeed less suitable for a modern spectrometer such as the one used in our work. To address the properties of the used MS instrumentation, we have now included a more suitable and more up to date reference (Nature methods, 2019, 16(7):587-594). We also slightly edited the footnote of Supplementary table 3 accordingly about the fitting rule of thumb for the QTOF MS instrumentation we used and updated the accuracy (20 ppm). It has to be noted that all our measurements are within the range of this accuracy and all the results observed are reproducible. Thus, this edition will not change the meaning of the sentence we presented.

*Supplementary Table 3: “*The accuracy of ESI-MS for intact protein molecules when using modern quadrupole time-of-flight (QTOF) MS has been documented to be 20 p.p.m when compared with the theoretical value³⁵.”*

Reference

35 Donnelly, D. P. et al. Best practices and benchmarks for intact protein analysis for top-down mass spectrometry. Nat. Methods. 16, 587-594 (2019).